# S-acylation by ZDHHC20 targets ORAI1 channels to lipid rafts for efficient Ca²⁺ signaling by Jurkat T cell receptors at the immune synapse

Amado Carreras-Sureda[1], Laurence Abrami[2], Kim Ji-Hee[3], Wen-An Wang[1], Christopher Henry[1], Maud Frieden[1], Monica Didier[1], F Gisou van der Goot[2], Nicolas Demaurex[1]*

[1]Department of Cell Physiology and Metabolism, Geneva, Switzerland; [2]Faculty of Life Sciences, Global Health Institute, Ecole Polytechnique Fédérale de Lausanne, Lausanne, Switzerland; [3]Department of Physiology, Yonsei University Wonju College of Medicine, Wonju, Republic of Korea

**Abstract** Efficient immune responses require Ca²⁺ fluxes across ORAI1 channels during engagement of T cell receptors (TCR) at the immune synapse (IS) between T cells and antigen presenting cells. Here, we show that ZDHHC20-mediated S-acylation of the ORAI1 channel at residue Cys143 promotes TCR recruitment and signaling at the IS. Cys143 mutations reduced ORAI1 currents and store-operated Ca²⁺ entry in HEK-293 cells and nearly abrogated long-lasting Ca²⁺ elevations, NFATC1 translocation, and IL-2 secretion evoked by TCR engagement in Jurkat T cells. The acylation-deficient channel remained in cholesterol-poor domains upon enforced ZDHHC20 expression and was recruited less efficiently to the IS along with actin and TCR. Our results establish S-acylation as a critical regulator of ORAI1 channel trafficking and function at the IS and reveal that ORAI1 S-acylation enhances TCR recruitment to the synapse.

*For correspondence:
nicolas.demaurex@unige.ch

Competing interest: The authors declare that no competing interests exist.

## Editor's evaluation

This study uses a wide range of approaches to identify acylation as a key regulator of Orai1 channel activation and calcium entry. The results show a novel role for Orai1 acylation in promoting signaling and early activation events upon interaction of T cells with antigen-presenting cells.

## Introduction

The development of efficient immune responses by T lymphocytes require long-lasting Ca²⁺ elevations mediated by the plasma membrane (PM) channel ORAI1 during engagement of T cell receptors (TCR) at the immune synapse (IS) forming between T cells and antigen-presenting cells (*Lioudyno et al., 2008*; *Cahalan and Chandy, 2009*; *Hartzell et al., 2016*). Following TCR engagement, the Ca²⁺ depletion of the endoplasmic reticulum (ER) causes the ER-bound Ca²⁺ sensors STIM1-2 to oligomerise and to accumulate in ER-PM junctions (*Zhang et al., 2005*; *Liou et al., 2005*), where they trap and gate the Ca²⁺-release-activated (CRAC) ORAI1 channel (*Vig et al., 2006*). The ensuing Ca²⁺ influx sustains long-lasting Ca²⁺ signals that initiate gene expression programs of T cell proliferation and differentiation. Proper ORAI1 function is essential for immunity in humans and patients with ORAI1 mutations suffer from severe combined immunodeficiency (*Lacruz and Feske, 2015*; *Feske et al., 2006*). Recent studies have revealed the structural rearrangements occurring within ORAI1 as STIM1

binding opens the channel pore (*Hou et al., 2018*; *Hou et al., 2020*) and increases its selectivity for Ca$^{2+}$ [11, 12], reviewed in *Qiu and Lewis, 2019*. Crystal structure from the highly homologous *Drosophila* Orai1 channel revealed a hexamer of four concentric TM subunits, with pore-lining TM1 helixes bearing an acidic selectivity filter followed by hydrophobic and basic regions (*Hou et al., 2018*; *Hou et al., 2012*). The closed structure is stabilised by multiple interactions between interlocking TM2 and TM3 helixes and peripheral TM4 helixes, bent in three crossed helical pairs extending in the cytosol. STIM1 binds to the external M4 helix, generating a gating signal transmitted by the TM2/TM3 ring to TM1, opening the channel pore and increasing its Ca$^{2+}$ selectivity (*Zhang et al., 2005*; *Liou et al., 2005*; *McNally et al., 2012*). The reversible switch of ORAI1 between a quiescent to an active state is highly regulated to avoid inappropriate Ca$^{2+}$ fluxes at the wrong time or place (reviewed in *Qiu and Lewis, 2019*).

Protein S-acylation, the reversible thioester linkage of a medium length fatty acid, often palmitic acid, on intracellular cysteine residues, dynamically controls the trafficking and gating of more than 50 ion channels by increasing the hydrophobicity of protein domains *Shipston, 2011*. S-acylation regulates ligand-gated (AMPA, GABA, Kainate, nAChR, NMDA, K$_{ATP}$, P2 × 7 receptors), voltage-gated (Ca$_V$, K$_V$, K$_{Ca}$ Na$_V$), epithelial (ENaC), and water channels (AQP4). The S-acylation reaction is mediated by zinc-finger and DHHC-domain containing Protein AcylTransferases (PATs) at the ER and Golgi *Ohno et al., 2006* and reversed by acyl protein thioesterases at the PM *Yokoi et al., 2016*, with 23 PATs and five thioesterases isoforms identified in human so far *Tabaczar et al., 2017*. Due to the hydrophobic nature of the attached acyl moieties, protein S-acylation impacts the distribution of proteins in membrane microdomains and between intracellular membranes *Rocks et al., 2010*. Indirect evidences suggest that ORAI1 activity might also be controlled by S-acylation. First, the human isoform ORAI1 was identified by acyl-biotinyl exchange chemistry coupled to mass spectrometry as a robustly S-acylated proteins in primary human T cells *Morrison et al., 2015* and human platelets *Dowal et al., 2011*. Second, the mouse Orai1 orthologue is reportedly S-acylated in neural stem cells and in a T-cell hybridoma *Martin et al., 2011*; *Li et al., 2012* according to the protein S-acylation database SwissPalm (https://swisspalm.org). Orai isoforms have two conserved Cys residues at potential S-acylation sites: Cys$^{143}$, located in a privileged S-acylation position at the edge of the second transmembrane domain (TM2), and Cys$^{126}$ within TM2. A third Cys residue, Cys$^{195}$, sensitive to oxidation *Bogeski et al., 2010*; *Alansary et al., 2016*, is exposed to the extracellular side and thus unlikely to be S-acylated. Among these three Cys residue, Cys$^{143}$ is the only one conserved in *C. elegans* (*Figure 1—figure supplement 1*).

Here, we show that the pore-forming subunit of the CRAC channel ORAI1 can undergo S-acylation at Cys$^{143}$ and that this modification is required for efficient channel activity and to promote recruitment of TCR at the immune synapse. Cys$^{143}$ but not Cys$^{126}$ substitutions prevent ORAI1 S-acylation mediated by PAT3, 7 and 20, with PAT20 being the only regulator of channel function. S-acylation-deficient ORAI1-C143A resides in cholesterol-poor membrane domains, accumulates to a lesser extent in STIM1 puncta upon store depletion, mediates reduced SOCE and I$_{CRAC}$, and is recruited less efficiently to the immune synapse in Jurkat T cells, severely impairing synapse formation and T cell activation. S-acylation of ORAI1 by PAT20 therefore controls the recruitment and function of channels and receptors at the immune synapse to mediate efficient T cell responses.

## Results

### The ORAI1 channel can undergo S-acylation on cysteine 143

Orai1 can potentially be S-acylated according to the SwissPalm 2.0 S-acylation database (https://swisspalm.org/) that compiles palmitoyl-proteomes *Blanc et al., 2015*. To validate that ORAI1 can undergo S-acylation, we assessed whether PEG-5k or tritiated palmitate could be incorporated by palmitoyl-thioester bonds into endogenous ORAI1 channels. HeLa cells were lysed in the presence of N-ethylmaleimide (NEM) to block free thiols, treated or not with hydroxylamine (HA) to break acyl-thioester bonds, and then with PEG-5k to label S-acylation sites. A mobility shift was observed in the presence of PEG-5k on western blots with anti-ORAI1 antibodies (*Figure 1A*). Tritiated palmitate was detected by autoradiography in HeLa cells labelled for 2 hr with $^3$H-palmitic acid and immunoprecipitated with anti-ORAI1 antibodies (*Figure 1B*). These data show that endogenous ORAI1 channels incorporate palmitic acid and can be labelled by acyl exchange of the acyl moiety with PEG-5k. A

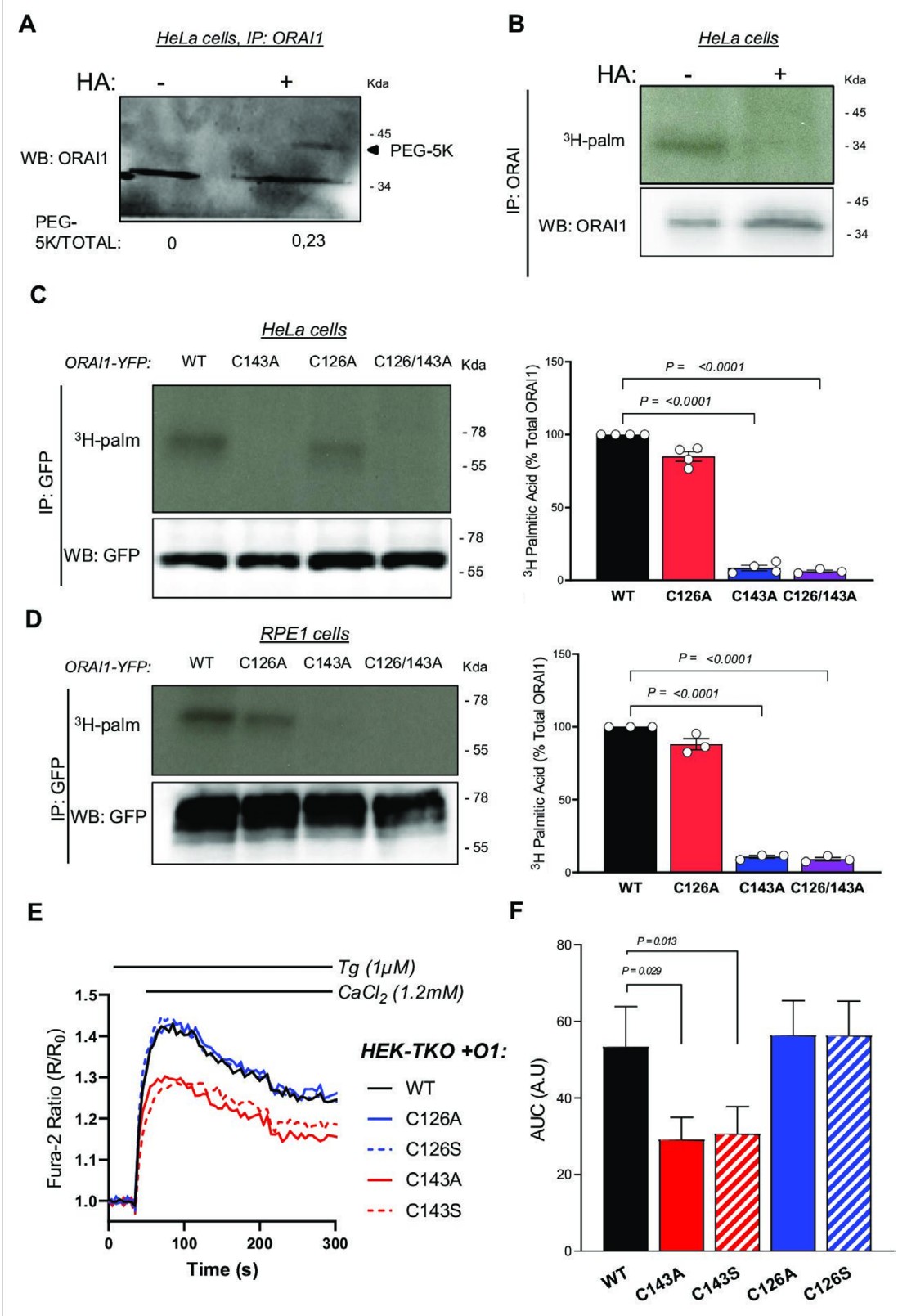

**Figure 1.** ORAI1 is S-acylated at cysteine C143. (**A**) ORAI1 immunoblot of HeLa cells treated with PEG-5k to label S-acylation sites after exposure to NEM to block free thiols and then to hydroxylamine (HA) to break acyl-thioester bonds. (**B**) Western blot and corresponding autoradiogram of HeLa cells labelled for 2 hr with $^3$H-palmitic acid with or without HA and immunoprecipitated with anti-ORAI1. (**C, D**) Western blots and corresponding autoradiograms of HeLa (**C**) and RPE-1 (**D**) cells expressing the indicated YFP-tagged ORAI1 mutants labelled with $^3$H-palmitic acid and

*Figure 1 continued on next page*

*Figure 1 continued*

immunoprecipitated with anti-GFP. Graph bars correspond to the mean ± SEM $^3$H-palmitic acid incorporation normalised to WT construct in three to four independent experiments. (**E**) Normalised mean fura-2 responses evoked by Ca$^{2+}$ readmission in HEK-TKO cells transiently transfected with the indicated ORAI1-YFP constructs and pre-exposed to Tg 1 μM for 8 min. (**F**) Area under the curve of the responses in E. Data are mean ± SEM of eight independent experiments. One-way ANOVA Dunnett's multiple comparisons test.

The online version of this article includes the following figure supplement(s) for figure 1:

**Figure supplement 1.** Right, Orai1 protein sequences aligned with CLustalW for the indicated organisms.

**Figure supplement 2.** Top; fluorescence images of WT O1/S1 cells (left) and averaged mCh-STIM1 and ORAI1-YFP fluorescence of the different O1/S1 stable cell lines Data are mean ± SE of 53–132 cells.

single band of higher molecular weight was observed in the acyl-PEG assay, indicating that a single residue of the ORAI1 channel can be S-acylated in these conditions. The fact that a non-shifted ORAI1 band remains indicates that only a sub-population undergoes S-acylation under our experimental conditions.

S-Acylation occurs on cysteine residues, present in ORAI1 at positions 126, 143 and 195, with C143 conserved up to *C. elegans* and C195 facing the extracellular side (*Figure 1—figure supplement 1*). To test whether C126 and/or C143 are S-acylation sites, we overexpressed ORAI1-YFP fusion proteins bearing substitutions at these residues in HeLa cells and evaluated $^3$H-palmitate incorporation by autoradiography. Cells expressing ORAI1-YFP bearing the C143A substitution or the double C126A/C143A mutation, but not the single C126A mutation, failed to incorporate $^3$H-palmitate (*Figure 1C*). Identical results were obtained with these ORAI1-YFP mutants expressed in RPE1 cells (*Figure 1D*), establishing that ORAI1 channels can undergo palmitoylation at residue C143.

## S-acylation potentiates ORAI1 channel function

S-acylation can alter ion channel trafficking, gating, and distribution in membrane lipids. To understand if S-acylation could affect ORAI1 activity, we measured Ca$^{2+}$ fluxes carried by ORAI1-YFP fusion constructs bearing substitutions at C126 and C143. In HEK-293 cells lacking all three ORAI isoforms (HEK-TKO, kindly provided by Rajesh Bhardwaj *Alansary et al., 2016*), transient expression of wild-type ORAI1 reconstituted Ca$^{2+}$ fluxes upon store depletion (*Figure 1E–F*). C143 substitutions by alanine or serine, but not C126 substitutions, reduced ORAI1-mediated SOCE (*Figure 1E–F*). These findings were confirmed by alanine substitutions in HEK-293 cells stably expressing mCherry-STIM1 (mCh-STIM1) and ORAI1-YFP, (HEK-S1/O1). Although these cell lines were sorted for the same fluorescence and presented comparable STIM1 and ORAI1 levels as judged by epifluorescence microscopy, SOCE responses were strongly reduced in cells bearing the C143A mutation (*Figure 1—figure supplement 2* and *Figure 2A–B*). We then recorded I$_{CRAC}$ currents in HEK-S1/O1-WT and -C143A cell lines and observed a current density reduction of fivefold in cells expressing the C143A mutant in cells perfused with 10 mM BAPTA (*Figure 2C–E*). The currents retained the inward rectification, positive reversal potential, and Gd$^{3+}$-sensitivity characteristic of highly Ca$^{2+}$ selective CRAC currents but activated more slowly in C143A cells (*Figure 2D–F , and Figure 2—figure supplement 1*). We then applied hyperpolarising pulses ranging from –120 to –60 mV to cells perfused with 10 mM EGTA to record fast Ca$^{2+}$-dependent inactivation (FCDI). C143A cells had reduced overall apparent FCDI, with a more rapid current inactivation followed by a delayed time-dependent potentiation that clipped off an initially normal FCDI (*Figure 2—figure supplement 2*). Our Ca$^{2+}$ imaging and electrophysiological data thus establish that replacing the S-acylated Cys 143 residue within ORAI1 reduces the CRAC channel function and alters its electrophysiological properties.

## S-acylation promotes ORAI1 clustering and PM mobility

To gain insight into the underlying mechanism, we evaluated whether S-Acylation impacts the expression or distribution of ORAI1 in the PM. We used a YFP-ORAI1-HA construct bearing the HA epitope in the extracellular loop between TM3 and TM4 (kindly provided by Khaled Machaca *Hodeify et al., 2015*). The HA immunoreactivity was comparable in non-permeabilised cells transiently expressing WT and C143A HA-tagged ORAI1, indicating similar surface expression (*Figure 3A*). Furthermore, the ratio of TIRF vs. total YFP fluorescence was comparable in HEK-S1/O1-WT and -C143A cells, confirming that the PM fraction of ORAI is not altered by the cysteine mutation (*Figure 3—figure*

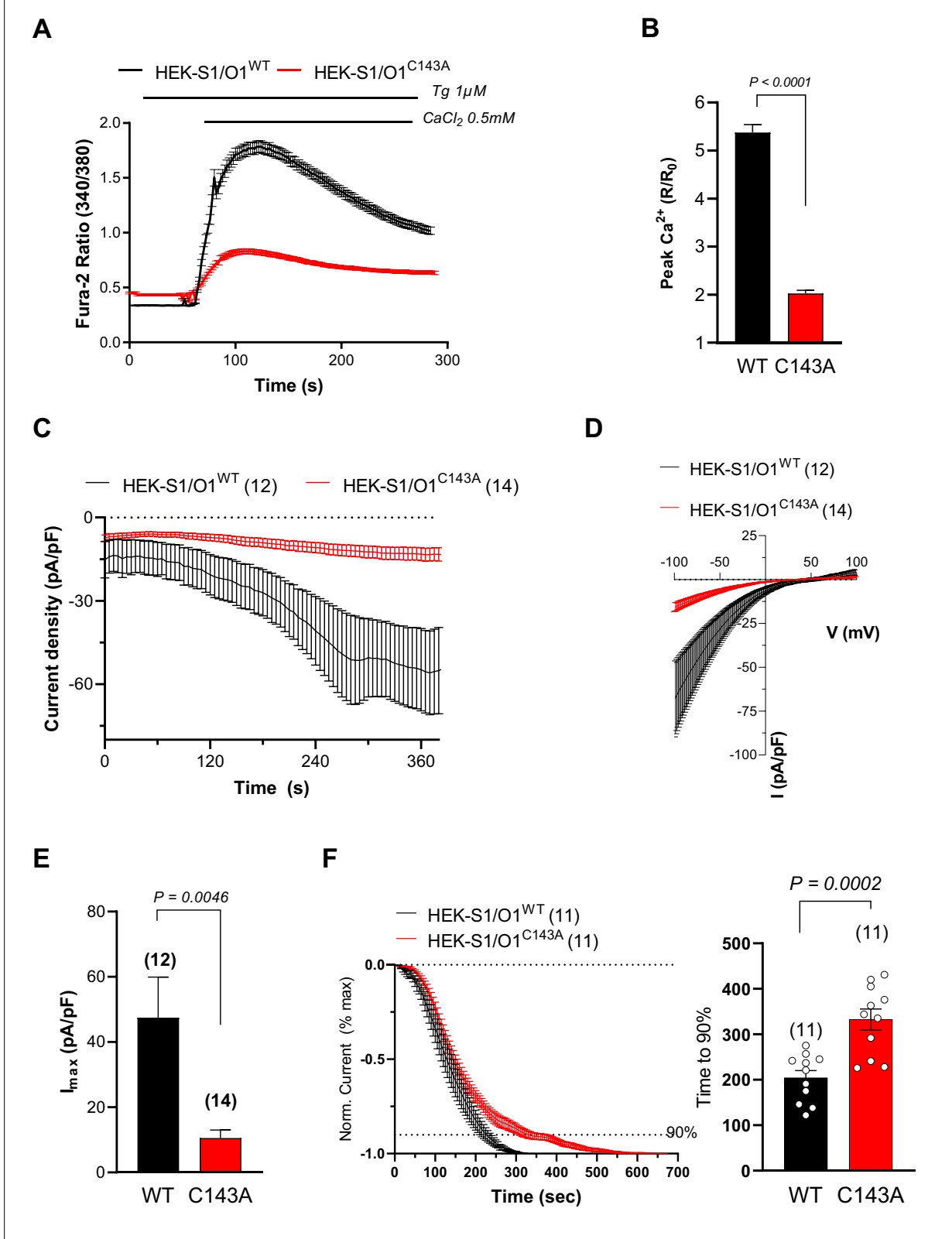

**Figure 2.** Preventing ORAI1 S-acylation reduces $I_{CRAC}$ currents. (**A**) Averaged fura-2 responses evoked by $Ca^{2+}$ readmission to Tg-treated (1 µM for 8 min) HEK-293 cells stably expressing mCh-STIM1 and ORAI1-YFP (O1/S1) bearing or not the C143A mutation. (**B**) Fold peak amplitude (Peak divided by basal) of the responses in A. Data are mean ± SEM of 196 (WT) and 198 (C143A) cells from five independent experiments. (**C**) Averaged $I_{CRAC}$ recordings of WT and C143A O1/S1 cells, measured every 5 sec at –100 mV. $I_{CRAC}$ was activated by cell dialysis with 10 mM BAPTA and 10 µM $Gd^{3+}$ added at the

*Figure 2 continued on next page*

*Figure 2 continued*

end of the recordings (see *Figure 2—figure supplement 1*). Data are mean ± SEM of 12 (WT) and 14 (C143A) cells. (**D**) Averaged current-voltage relationship of steady-state $Gd^{3+}$-sensitive currents in the cells in C. (**E**) Peak current densities ($I_{max}$) of WT and C143A O1/S1 cells after subtraction of basal or $Gd^{3+}$-insensitive currents (same cells as C). (**F**) Time-course of current activation. Left: Averaged recordings with basal and maximal values set to 0 and –1, respectively, aligned at t = 0 for break-in. Cells with pre-activated currents were not included. Right: Statistical evaluation of the activation time. Data are mean ± SEM, of 11 (WT and C143A) cells. Two-tailed unpaired Student's *t*-test.

The online version of this article includes the following figure supplement(s) for figure 2:

**Figure supplement 1.** Representative $I_{CRAC}$ recordings of WT and C143A O1/S1 HEK-293 cells.

**Figure supplement 2.** Families of currents evoked by 200 ms voltage steps to voltages ranging from –120 to –60 mV in steps of 20 mV in WT and C143A O1/S1 perfused with 10 mM EGTA to enable FCDI.

*supplement 1*). We then recorded the formation of STIM1 and ORAI1 clusters in HEK-S1/O1 cells during store depletion by TIRF microscopy. The C143A mutation reduced the number of ORAI1 clusters and the average ORAI1-YFP fluorescence accumulating in STIM1-containing puncta as well as the co-localisation between mCh-STIM1 and ORAI1-YFP while the kinetics parameters of mCh-STIM1 and ORAI1-YFP cluster formation were not altered by the mutation (*Figure 3B–E and Figure 3—figure supplement 2*). Lipid incorporation into proteins changes their lipophilic preference, and potentially their membrane mobility. To assess ORAI1 mobility in the PM, we used fluorescence recovery after photobleaching (FRAP) and measured the lateral diffusion of ORAI1-YFP in fluorescent matched HEK-S1/O1-WT and -C143A cell lines. The C143A mutant had a significantly lower diffusion coefficient indicative of a reduced mobility in membrane lipids (*Figure 3—figure supplement 3* S3E). These results indicate that the acylation-deficient ORAI1-C143A has a reduced mobility in the PM and is poorly recruited to ER-PM junctions upon store depletion.

## Protein *S*-acyl transferase 20 (PAT20) mediates ORAI1 S-acylation and regulates channel function

S-Acylation is exerted by DHHC-domain containing protein acyltransferases (PATs) proteins, which form a large family of enzymes containing 23 members. To identify the enzyme(s) promoting ORAI1 S-acylation, we transiently transfected ORAI1-YFP in RPE1 cells stably expressing different PATs and measured palmitate incorporation in the immunoprecipitated fraction by autoradiography. An enhanced palmitate incorporation was observed in RPE1 and Hela cells expressing PAT3, PAT7 and PAT20 (*Figure 4A*). Two bands migrating at ~50–65 kDa were labelled in cells overexpressing PAT3, 7 and 20, corresponding to the two ORAI1 forms generated by alternative translation initiation, fused to YFP. We then assessed whether the enhanced ORAI1 S-acylation conferred by PATs overexpression could modulate channel activity. Enforced expression of PAT20 but not of PAT3 or PAT7 in HeLa cells enhanced SOCE (*Figure 4B*). All siRNAs decreased SOCE equally in WT but not in C143A HEK S1/O1 cells but only siRNAs directed against PAT20 were specific (*Figure 4—figure supplement 1*). Importantly, PAT20 silencing mimicked the effect of the C143A mutation when an identical $Ca^{2+}$ concentration was reapplied, linking the SOCE defect to the lack of ORAI1 S-acylation (*Figure 4—figure supplement 2*). Accordingly, PAT20 potentiated SOCE in HEK-293 cells stably expressing ORAI1-WT but not the C143A mutant (*Figure 4C*), indicating that the gain of function conferred by increased PAT20-driven S-acylation requires this cysteine residue. To test whether the increased S-acylation impacts ORAI1 lipid partitioning, we generated giant plasma membrane vesicles (GPMV) from HEK-293 cells transiently expressing ORAI1-YFP constructs and PAT20 and measured the lipid distribution of the WT and mutated channel using cholera toxin B and the PH-Domain of PLC-Ɣ (PIP2-Cherry) as lipid raft and non-raft markers. PAT20 overexpression increased the fraction of WT ORAI1-YFP located in lipid rafts, without altering the distribution of the C143A channel (*Figure 4D*). These experiments indicate that PAT3, 7 and 20 S-acylate ORAI1 on a single residue, Cys143, and that PAT20 levels limit channel function. PAT20-mediated S-acylation of Cys143 modifies the lipid preference of the ORAI1 channel to promote its accumulation in lipid rafts.

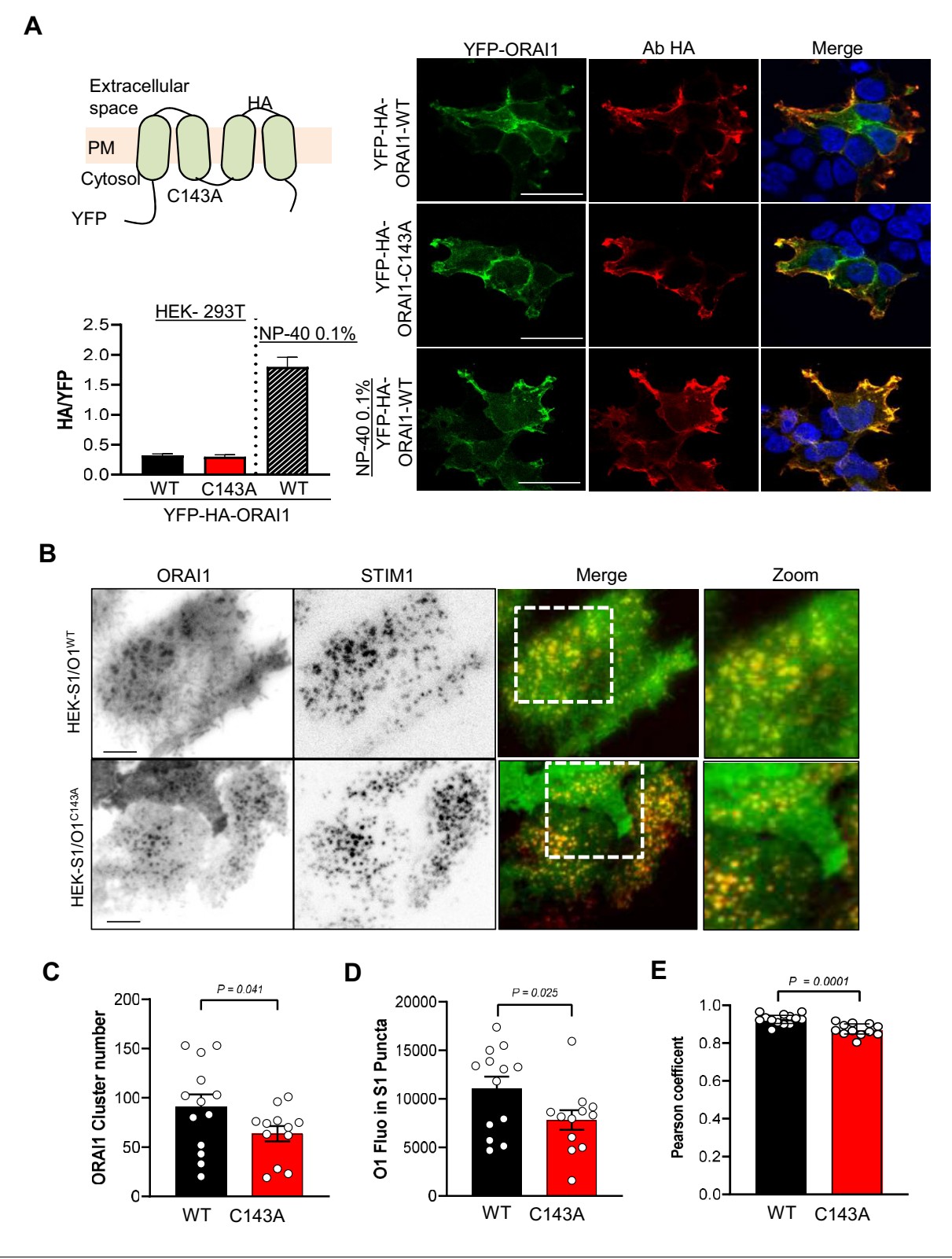

**Figure 3.** Preventing ORAI1 S-acylation reduces channel clustering and affinity for lipid rafts. (**A**) Cartoon of the YFP-HA-ORAI WT or C143A constructs with HA exposed extracellularly between TM3 and TM4. HEK-293 cells transiently expressing the indicated constructs were stained for HA in non-permeabilising conditions to decorate the ORAI1 PM fraction. Graph bars represent the mean ± SEM of 15 (WT), 17 (C143A) and 10 (WT+ NP-40) cells. Right, Representative confocal images of YFP (green) and HA (red) staining in the indicated conditions. Scale bar = 25 μm. (**B**) Representative

*Figure 3 continued on next page*

Figure 3 continued

TIRF images of WT and C143 O1/S1 cells exposed to 10 μM CPA for 10 min to induce mCh-STIM1 and ORAI1-YFP clustering. Bars = 5 μm. (C–E) Quantification of ORAI1-YFP clusters numbers (C), of the average ORAI1-YFP fluorescence inside STIM1 clusters (D), and of the Pearson's co-localisation coefficient between mCh-STIM1 and ORAI1-YFP (E) 10 min after CPA exposure. Data are mean ± SEM of 13 (WT) and 12 (C143A) cells from three independent experiments. (C, E) Two-tailed unpaired Student's *t*-test. (D) One tailed.

The online version of this article includes the following figure supplement(s) for figure 3:

**Figure supplement 1.** Fractional TIRF vs. total ORAI1-YFP fluorescence of WT and C143A O1/S1 cells.

**Figure supplement 2.** Time-course of CPA-induced changes in ORAI1-YFP (top) and mCh-STIM1 (bottom) fluorescence (left), number of clusters (middle), and relative cluster area (right, with minimal value set to 0 and maximal value to 100).

**Figure supplement 3.** FRAP recordings of WT and C143 O1/S1 cells.

## ORAI1 S-acylation is required for TCR-mediated long-lasting Ca²⁺ elevations in Jurkat T cells

To assess whether ORAI1 S-acylation impacts T cell function, we generated by CRISPR an ORAI1-deficient Jurkat T cell line, in which SOCE was severely blunted (*Figure 5A* and *Figure 5—figure supplement 1*). Stable transduction of ORAI1-WT restored SOCE in these cells while ORAI1-C143A, expressed at comparable levels at the PM, was ineffective (*Figure 5A–B* and *Figure 5—figure supplement 2*). Activation of the TCR with CD3/CD28 beads evoked long-lasting Ca²⁺ elevations in CRISPR ORAI1+ ORAI1 WT stable cells. In contrast, cells reconstituted with ORAI1-C143A exhibited smaller and more transient responses (*Figure 5C* and *Figure 5—figure supplement 3*). Among all 23 PAT isoforms overexpression of only PAT 3, 7, and 20 increased palmitate incorporation in ORAI1-deficient Jurkat T cells reconstituted with ORAI1-WT but not with the C143A mutant, confirming that ORAI1 is S-acylated by these enzymes in immune cells (*Figure 5—figure supplement 4*). Accordingly, PAT20 expression augmented SOCE in WT cells but had no effect in cells lacking ORAI1 (*Figure 5—figure supplement 5*). We then tested whether the downstream responses of T cells were similarly affected by measuring NFAT endogenous translocation, NFAT-dependent transcription, and IL-2 production. ORAI1 ablation aborted NFAT-dependent transcription and reduced nuclear translocation of endogenous NFATC1 in cells treated with Tg (*Figure 5D–E*) and prevented the potentiating effects of PAT20 expression on IL-2 production evoked by Tg and CD3-coated plates (*Figure 5—figure supplement 5*). Compared to C143A, re-expression of WT ORAI1 potentiated NFAT-dependent transcription and NFATC1 nuclear translocation in ORAI1-deficient Jurkat cells treated with Tg (*Figure 5D–E , and Figure 5—figure supplement 6*). Similar findings were obtained with CD3-coated plates, which increased NFATC1 translocation and IL-2 production in cells reconstituted with WT but not with C143A ORAI1 (*Figure 5F and G*). Furthermore, PAT20 silencing reduced IL-2 production by WT but not C143A Jurkat T cells and mimicked the effect of the C143A mutation, linking the defect to the lack of ORAI1 S-acylation (*Figure 5—figure supplement 7*). These results indicate that ORAI1 S-acylation at C143A is required for efficient activation of Jurkat T cells following TCR engagement.

## ORAI1 S-acylation sustains TCR recruitment and signaling at the immune synapse

TCR activation triggers the formation of an immune synapse (IS) between T cells and antigen-presenting cells, a structure that maximises the membrane contact area and organises TCR and signalling proteins into concentric zones *Lioudyno et al., 2008*; *Bromley et al., 2001*. ORAI1 channels are rapidly recruited into the IS *Lioudyno et al., 2008*; *Barr et al., 2008* and are required for the formation of dynamic actin structures *Hartzell et al., 2016* in a self-organising process enabling long-lasting local Ca²⁺ signals to initiate gene expression programs that drive T cell proliferation *Feske et al., 2001*. To test whether S-acylation impacts IS formation and downstream signalling, we co-cultured ORAI1-deficient cells reconstituted with WT or mutant ORAI1-YFP with Staphylococcus enterotoxin E (SEE)-pulsed Raji cells and evaluated T cell activation and IS formation in this IS model *Chichili et al., 2010*; *Bello-Gamboa et al., 2019* (*Figure 6A*). After 24 hr of co-culture, a strong NFAT luciferase activity was observed in Jurkat T cells reconstituted with ORAI1-WT, while no response was observed in cells reconstituted with ORAI1-C143A (*Figure 6B*). We then quantified ORAI1 accumulation in the IS by live imaging using SEE-pulsed Raji labelled with the PM stain CellMask. WT ORAI1-YFP accumulated in the IS in 86 % of cells imaged, consistent with previous reports (*Lioudyno et al., 2008*)

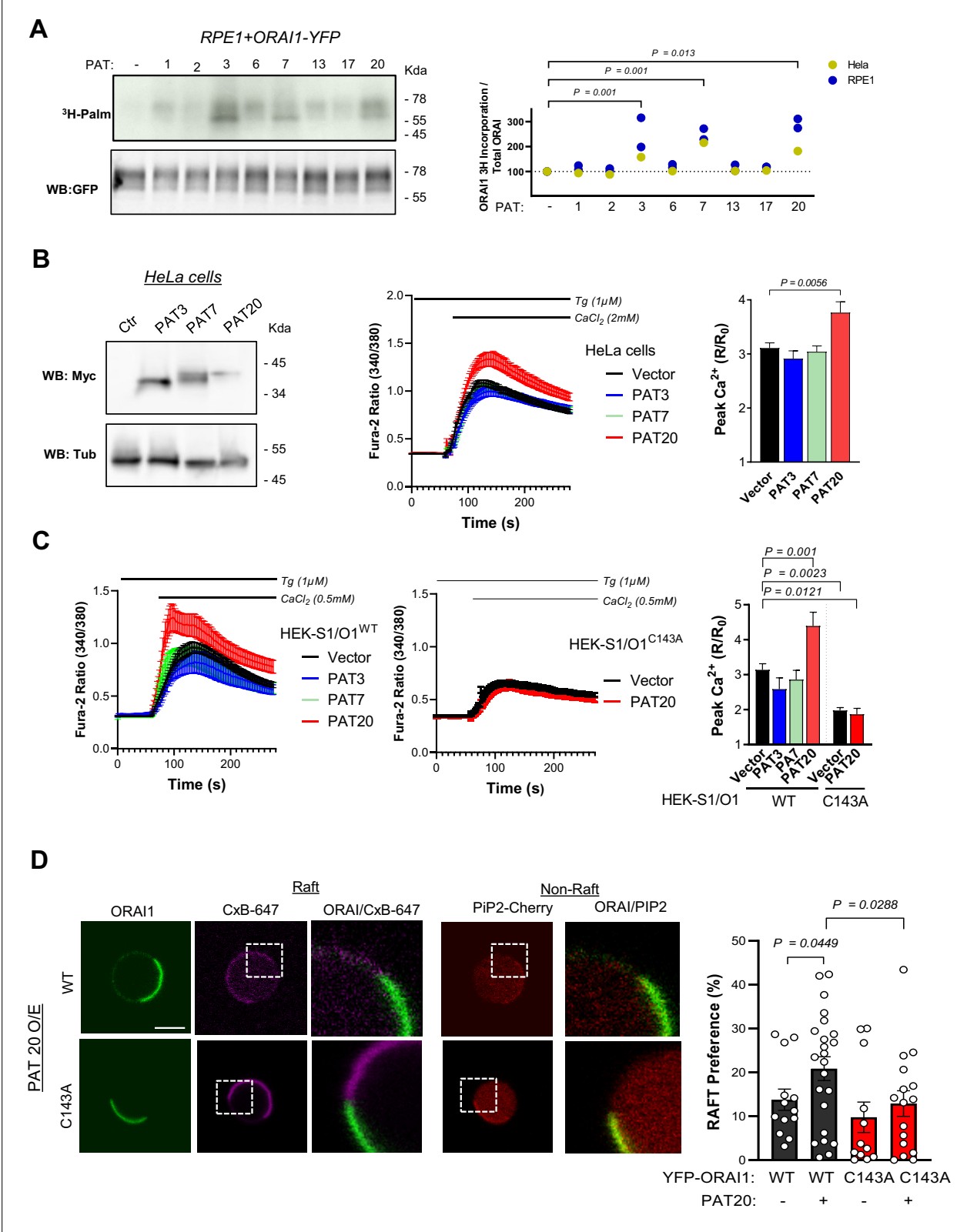

**Figure 4.** PAT20 S-acylates ORAI1 and modulates its activity. (**A**) Western blot and matching autoradiogram of RPE-1 cells expressing ORAI1-YFP plus the indicated PAT isoform, labelled with ³H-palmitic acid and immunoprecipitated with anti-GFP. Graph bar (right) shows densitometry analysis of the tritiated bands relative to GFP in RPE1 (Blue, N = 2) and HeLa cells (yellow, N = 1). (**B**) Functional effect of PAT3, 7, and 20 expression. Representative western blot of HeLa cells expressing Myc-tagged PAT isoforms (left), averaged SOCE responses (middle), and peak amplitude (right). Data are mean

*Figure 4 continued on next page*

*Figure 4 continued*

± SEM of 49–74 cells from three independent experiments. (**C**) Averaged SOCE responses of WT (left) or C143A (middle) S1/O1 cells expressing these PAT isoforms and their peak amplitude (right). Data are mean ± SEM of 31–129 cells from five independent experiments. (**D**) Lipid partitioning of ORAI1 in giant plasma membrane vesicles from HEK-293 cells transiently transfected with PiP2-Cherry and WT or C143 ORAI1-YFP together with PAT20 or PCDNA3. Left: representative fluorescence images of vesicles from cells expressing PAT20 and WT or C143 ORAI1-YFP (green) stained with cholera toxin subunit B (Magenta) as raft marker and PiP2-Cherry (Red) as non-raft marker (scale bar = 2.5 µm). Right: Graph bar representing the % of ORAI1 preference for raft domains. Data are mean ± SEM of 13 (WT empty), 12 (C143A empty), 23 (WT+ PAT20), and 16 (C143A + PAT20) vesicles from four independent experiments. (**A–C**) One-way ANOVA Dunnett's multiple comparisons test. (**D**) One-tailed unpaired Student's *t*-test.

The online version of this article includes the following figure supplement(s) for figure 4:

**Figure supplement 1.** Top graph bars represent the PAT3, 7, and 20 mRNA expression levels in cells transfected with the indicated siRNA (N = 6–12 biological replicates in two to four independent transfections).

**Figure supplement 2.** Averaged SOCE responses (left), and peak amplitude (right) of HEK-TKO cells transiently transfected with the indicated siRNAs plus mCh-STIM1 and either WT or C143A ORAI1-YFP.

while C143A ORAI1-YFP was enriched in the opposite pole, forming distal caps, in nearly 60 % of cells (*Figure 6C–D and Figure 6—figure supplement 1*, *Figure 6—videos 1 and 2*). Kymographs of lines spanning the IS-distal cap axis revealed that WT ORAI1-YFP fluorescence was enriched 1.5-fold in the IS within 20 min of contact, while C143A ORAI1-YFP remained evenly distributed between the IS and distal caps (*Figure 6E* and *Figure 6—figure supplement 1*). To better visualise the molecular organisation of the IS, we performed confocal imaging of cells stained for actin and TCR. After 1 hr of co-culture, quantification of the IS-distal cap fluorescence profiles revealed that ORAI1-YFP, phalloidin, and TCR-PE levels were decreased in IS forming in Jurkat T cells reconstituted with the C143A mutant despite similar TCR plasma membrane levels (*Figure 6F–G , and Figure 6—figure supplement 2*). Finally, ERK1/2 phosphorylation was reduced in C143A cells plated on CD3-coated coverslips, indicating that TCR signalling is compromised in these cells (*Figure 6—figure supplement 3*). These data indicate that ORAI1 S-acylation enhances the recruitment and signalling of the ORAI1 channel and of TCR at the immune synapse in Jurkat T cells (*Figure 6H*).

## Discussion

In this study, we show that S-acylation of ORAI1 at a single cysteine residue enhances the affinity of the channel for cholesterol-rich lipid microdomains and promotes its trapping at the immune synapse, thereby enabling the local $Ca^{2+}$ fluxes that control the activation of Jurkat T cells. Using acyl-PEG exchange, palmitate incorporation, and mutagenesis, we show that ORAI1 can be chemically modified by S-acylation and identify the acylation site as Cys143 on the cytosolic rim of the second TM domain. Substitutions at Cys143 but not at Cys126 within TM2 prevented palmitate incorporation and decreased SOCE as well as $I_{CRAC}$. A comparable inhibition was observed with cysteine-less ORAI1 in an earlier study focusing on Cys195 substitutions that prevent $I_{CRAC}$ inhibition by hydrogen peroxide *Bogeski et al., 2010*. These data indicate that the ORAI1 channel is S-acylated at Cys143 and that replacement of this residue, but not of the two other ORAI1 cysteines, prevents S-acylation and impacts channel function, consistent with the recent report that Orai1 is rapidly S-acylated at cysteine 143 upon ER $Ca^{2+}$ store depletion *West et al., 2022*. Cys143 is the only cysteine conserved in all human isoforms and in ORAI1 homologs up to *C. elegans*, suggesting that S-acylation at this site is an evolutionary conserved function.

Using $Ca^{2+}$ imaging and electrophysiology, we establish that ORAI1 S-acylation has a significant functional impact on the channel function. Substitutions at Cys143, but not at Cys126, decreased SOCE by 50 % in HEK-293 cells when the channel was transiently expressed alone and by 80 % when it was stably co-expressed with STIM1. SOCE was also reduced when the S-acylation-defective ORAI1-C143A was expressed in HEK-293 cells lacking all ORAI isoforms or in Jurkat T cells lacking ORAI1, firmly linking the SOCE defect to the ORAI1 Cys143 mutation. Patch-clamp recordings confirmed that $I_{CRAC}$ currents were reduced by 80 % by the mutation when ORAI1-YFP was stably expressed together with mCh-STIM1, at identical expression levels. ORAI1-C143A currents retained the characteristic inward rectification and high $Ca^{2+}$ selectivity of CRAC channels but activated more slowly in cells perfused with 10 mM BATPA and exhibited a delayed time-dependent potentiation that masked an apparently normal FCDI. The slow activation might reflect a reduced $Ca^{2+}$-dependent inactivation

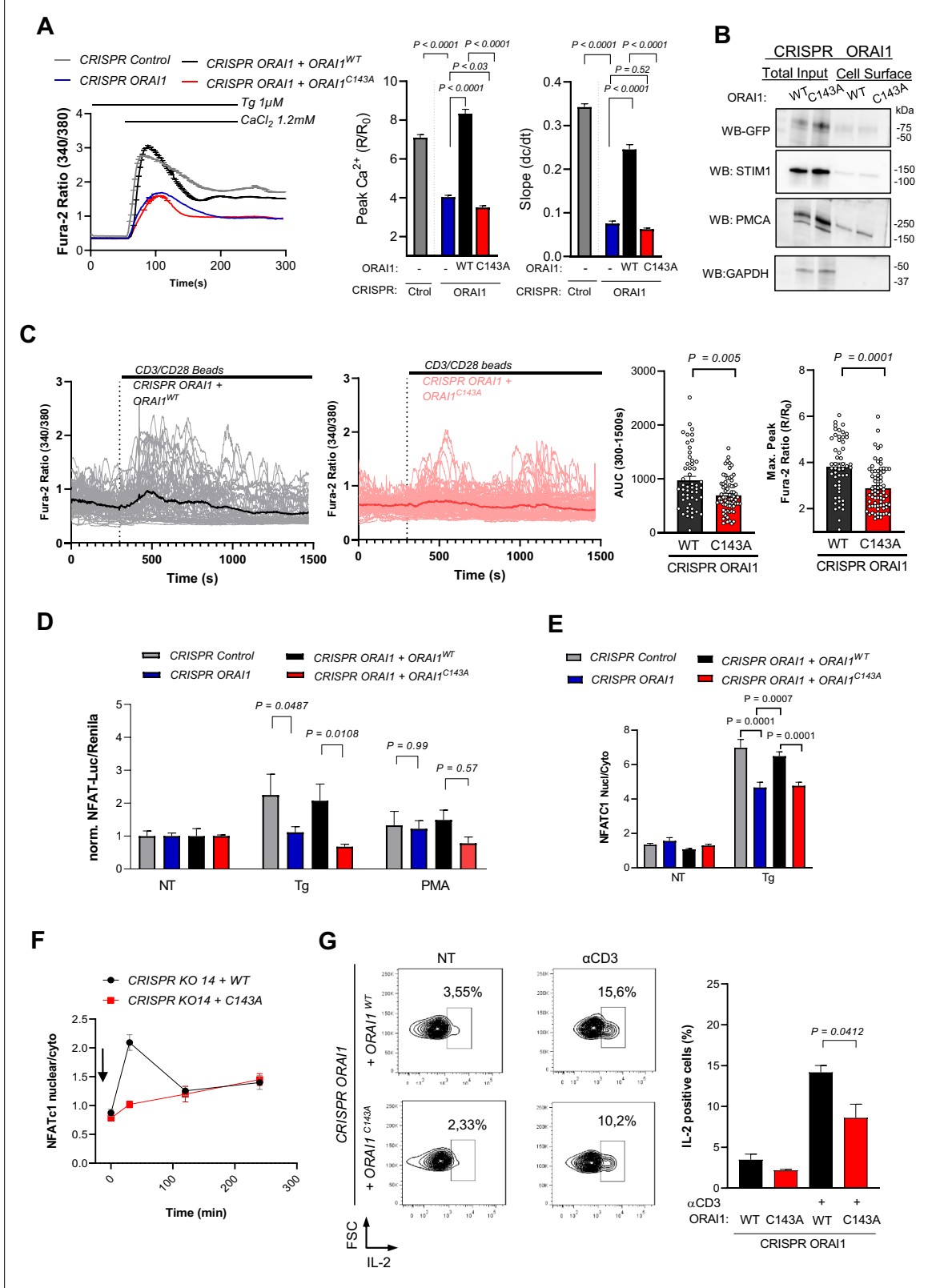

**Figure 5.** ORAI1 S-acylation promotes Jurkat T cell activation. (**A**) Averaged fura-2 responses, their peak amplitude (middle graph bar) and slope (right graph bar) evoked by Ca²⁺ re-addition in Tg treated (1 μM, 8 min) Jurkat cells lines generated by CRISPR with control or ORAI1-targeted guiding sequences and stably re-expressing either WT or C143A ORAI1-YFP. Data are mean ± SEM of 210 (Control), 242 (KO), 189 (WT), and 203 (C143A) cells from three independent experiments (**B**) Western blot showing the amount of biotinylated GFP immunoreactivity in the PM in Jurkat CRISPR ORAI1

*Figure 5 continued on next page*

*Figure 5 continued*

cells reconstituted with WT or C143A ORAI1-YFP. Representative of 2 independent experiments. (**C**) Individual (thin lines) and averaged (thick line) fura-2 recordings of Jurkat CRISPR ORAI1 cells reconstituted with WT or C143A ORAI1-YFP, exposed to CD3/CD28-coated beads in $Ca^{2+}$ containing solution (left). Averaged peak and integrated responses evoked by CD3/CD28 beads in individual cells during the recording period (right). Data are from 52 cells (WT) and 72 cells (C143A) from three independent experiments. (**D**) Relative changes in NFATC-Luciferase vs. housekeeping- Renilla luminescence evoked in 4 h by Tg (1 µM) and PMA (100 nM) in the indicated cell lines. Data are mean ± SEM of 8–12 biological replicates from three independent experiments. (**E**) Endogenous NFATC1 translocation evoked in 4 hr by Tg (1 µM) in the indicated cell lines, measured by immunofluorescence. Data are mean ± SEM of the nuclear to cytosol NFATC1 intensity ratio of 58–161 cells from four independent experiments. (**F**) Time-course of NFATC1 translocation evoked by plates coated with CD3 (OKT3 1 µg/ml). Data are mean ± SEM of 86–161 cells from five independent experiments. (**G**) IL-2 production evoked by untreated (NT) or surface coated CD3 (OKT3 1 µg/ml). Left, Representative density dot plots of Jurkat cells stained for IL-2. Graph bars represent the mean ± SEM of three independent experiments. One-way ANOVA Dunnett's multiple comparisons test (**A**), Sidak multiple comparisons test (**D and E**), two-tailed unpaired Student's *t*-test (C and G).

The online version of this article includes the following figure supplement(s) for figure 5:

**Figure supplement 1.** Sequences of genomic DNA used to generate the CRISPR ORAI1 Jurkat T cell lines (top) and FLAG or ORAI1 immunoblot of Jurkat T cells expressing FLAG-tagged Cas9 (bottom), WB are representative of two independent experiments.

**Figure supplement 2.** Fluorescence intensity profiles of CRISPR ORAI1 cells reconstituted with WT and C143 ORAI1-YFP measured by flow cytometry (N = 4).

**Figure supplement 3.** Representative Fura2 recordings of the responses evoked by CD3/CD28-coated beads in Jurkat CRISPR ORAI1 cells reconstituted with WT or C143A ORAI1-YFP (from Fig.*Figure 5C*).

**Figure supplement 4.** Top left blot indicates the[3]H-palmitate incorporation in ORAI1-deficient Jurkat T cells reconstituted with WT ORAI1-YFP transiently expressing the indicated PAT isoforms.

**Figure supplement 5.** Averaged SOCE responses (left) and peak SOCE amplitude (middle) measured with YC3.6 in WT or ORAI1-deficient Jurkat T cells expressing PAT20 or a control plasmid.

**Figure supplement 6.** NFATC1 immunoreactivity of ORAI1-deficient Jurkat T cells reconstituted with WT and C143 ORAI1-YFP treated with Tg 1 µM for 4 hr to induce nuclear translocation of NFATC1.

**Figure supplement 7.** Fraction of IL-2 positive ORAI1-deficient Jurkat T cells reconstituted with WT or C143A ORAI1-YFP and transfected with si-Scr and si-PAT20 adhered 24 hr on plates coated with CD3 (1 µg/ml).

given the low amplitude of ORAI1-C143A currents. A delayed time-dependent potentiation occurs at STIM:Orai ratios below 1:2 *Scrimgeour et al., 2009*, which is unlikely to occur in our conditions because we used a twofold higher amount of STIM1 plasmid in the transfection mixture to ensure a high STIM:Orai ratio. The ORAI1-C143A currents also had a positive reversal potential, indicative of a high STIM:Orai ratio *McNally et al., 2012*. The deacylated channel thus has distinct biophysical properties that might arise from the lack of acylation or from the C143A mutation itself impacting STIM1 binding or the conformational transitions occurring during channel gating and inactivation.

We further identify the zinc-finger and DHHC-containing S-acyltransferase zDHHC3, 7 and 20 (PAT3, 7 and 20) as mediating the S-acylation reaction. Among all 23 PAT family members exogenously expressed, only these three enzymes reproducibly increased the incorporation of tritiated palmitate into full-length ORAI1-YFP expressed in HeLa, RPE1, and Jurkat T cells. The three enzymes acylate a single residue, Cys143, but PAT20 was the only isoform that was consistently associated with a gain or loss of channel function. This robust effect was observed with both endogenous and exogenously expressed ORAI1 channels and required Cys143. These data suggest that ORAI1 undergoes successive cycles of S-acylation at Cys143, involving PATs 3, 7 and 20, and of de-acylation, with the three isoforms acting at different locations during the channel life cycle. PAT3 and PAT7 accumulate in the Golgi and PAT20 in the PM *Ohno et al., 2006*. We therefore propose that S-acylation in the Golgi is functionally neutral while S-acylation by PAT20 at the PM enhances channel function by impacting ORAI1 distribution in PM lipids. Our observation that PAT20 enhances the channel preference for cholesterol-rich domains of vesicles retrieved from the PM is consistent with this mechanism and with the enzyme acting at the PM.

Using TIRF imaging we then show that mutating the Cys143 S-acylation site reduces the amounts of channels recruited to ER-PM junctions during SOCE, without altering the kinetics of cluster formation. ORAI1 PM clusters are the macroscopic signature of ORAI1 trapping by STIM1, a dynamic event involving the entry and exit of ORAI1 particles into PM domains facing STIM1 molecules on apposed cortical ER cisternae *Wu et al., 2014*. The reduced presence of ORAI1-C143A in puncta is consistent with a recent report that S-acylation is required for the recruitment of active ORAI1 channels to STIM1

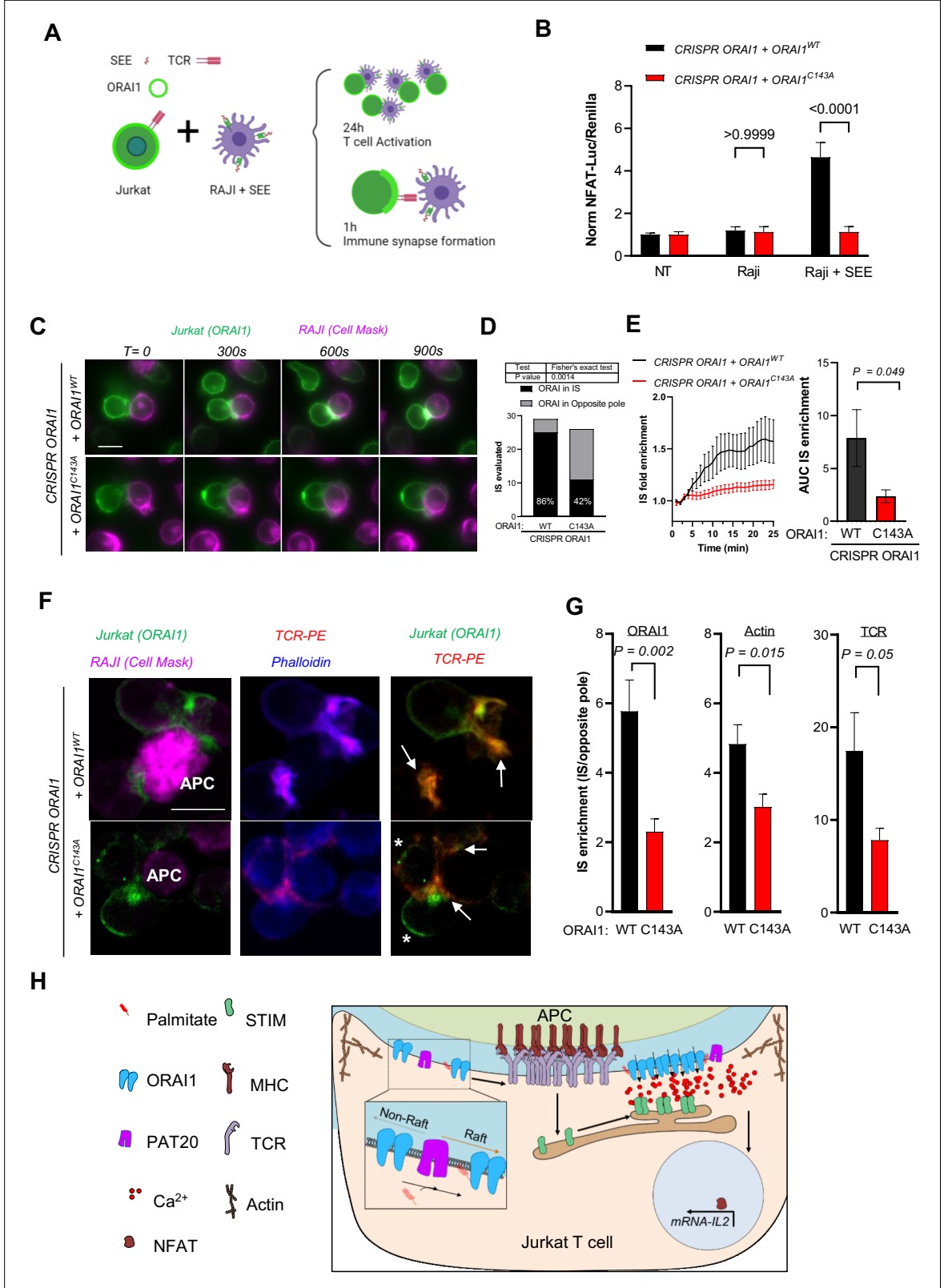

**Figure 6.** ORAI1 S-acylation regulates TCR enrichment and signaling at the immune synapse. (**A**) Conditions used to study synapse formation between Raji and Jurkat T cells. Raji are pulsed with SEE prior to co-culture with Jurkat cells, IS formation evaluated in living (25 min) and fixed cells (1 H) and Jurkat activation at 24 hr. (**B**) Relative changes in NFAT-Luciferase vs. housekeeping- Renilla luminescence evoked by coculture of the indicated cell lines for 24 hr with naïve or SEE (1 μg/ml) pulsed Raji. Data are mean ± SEM of 15–20 biological replicates in cells from four independent experiments.

*Figure 6 continued on next page*

*Figure 6 continued*

(**C**) Representative images of ORAI1-deficient Jurkat T cells reconstituted with WT or mutant ORAI1-YFP co-cultured with pulsed RAJI cells stained with CellMask deep red. (Scale bar = 10 μm). (**D**) ORAI distribution in IS vs. opposite pole in IS forming between SEE-pulsed Raji and ORAI1-deficient Jurkat T cells reconstituted with WT (29 IS) or mutant (26 IS) ORAI1. Chi-square p value: 0.0014, two-sided Fisher's exact test. (**E**) Quantification of ORAI1 enrichment in time-lapse images from C. IS accumulation was measured using kymographs of 20 pixels wide lines spanning the IS-distal cap axis, drawn on stable (not moving) IS forming in WT (12) and C143A (8) cells from three independent experiments. Graph bar shows mean ± SEM area under the curve of the kinetic enrichment graphs. (**F**) Representative confocal images of IS forming between SEE-pulsed Raji labeled with CellMask (Magenta, APC white labelling) and ORAI1-deficient Jurkat T cells reconstituted with WT or mutant ORAI1-YFP (Green), labelled with Phalloidin (Blue) and anti-TCR-PE (Red). Scale bar = 10 μm. Arrows indicate IS formation and asterisks accumulation of ORAI1 in the opposite pole. (**G**) IS enrichment for ORAI1-YFP, Phalloidin and TCR fluorescence in Jurkat T cells reconstituted with WT or C143A ORAI1-YFP. Fluorescence levels at the IS and opposite pole were extracted from 10 pixel wide line profiles. Graph bars shows mean ± SEM in WT (n = 25 IS) and C143A (n = 20 IS). Two-tailed unpaired Student's *t*-test (**B, E and G**); Chi square (**D**). (**H**) Proposed model: S-acylation by PAT20 targets ORAI1 to lipid rafts, enabling the coordinated recruitment of ORAI1 and TCR to the immune synapse for efficient signalling.

The online version of this article includes the following video and figure supplement(s) for figure 6:

**Figure supplement 1.** Top; Representative images of IS formation in ORAI1-deficient Jurkat T cells reconstituted with WT or C143A ORAI1-YFP, related to *Figure 6C*.

**Figure supplement 2.** Surface TCR-PE expression in the indicated cell lines determined by flow-cytometry.

**Figure supplement 3.** Phosphorylated vs.total ERK1/2 immunoreactivity of ORAI1-deficient Jurkat T cells reconstituted with WT or C143A ORAI1-YFP adhered or not for 15 min on plates coated with CD3 (1 μg/ml).

**Figure 6—video 1.** Related to Figure 6C–E.

https://elifesciences.org/articles/72051/figures#fig6video1

**Figure 6—video 2.** Related to Figure 6C–E.

https://elifesciences.org/articles/72051/figures#fig6video2

clusters *West et al., 2022*. Reduced ORAI1 trapping by STIM1 would reduce CRAC currents and possibly FCDI by increasing the amount of STIM1 molecules required to trap and gate deacylated channels. We show here that Cys143 substitution reduces ORAI1 mobility in lipids while enforced S-acylation promotes ORAI1 accumulation in ordered lipid domains rich in cholesterol. This suggests that S-acylation promotes channel trapping by increasing its lateral mobility in lipids. Alternatively, S-acylation could increase the affinity of ORAI1 for STIM1 or promote lateral interactions between channels to stabilise and enlarge ORAI1 clusters. Future experiments using single-molecule imaging are required to establish the underlying mechanism.

By re-expressing the acylation-resistant ORAI1-C143A in ORAI1-deficient Jurkat T cells, we show that ORAI1 S-acylation is required for the efficient activation of Jurkat T cells during TCR engagement by either CD3-coated coverslips, beads or SEE-pulsed Raji cells. Replacing the single ORAI1 S-acylation site strongly reduced SOCE and the long-lasting $Ca^{2+}$ elevations driven by TCR engagement and the ensuing NFATC1 translocation and IL-2 production, signature markers of T cell activation. Expressing PAT20 increased $Ca^{2+}$ responses and TCR-induced IL-2 synthesis in WT but not in ORAI1-deficient Jurkat T cells, confirming that ORAI1 S-acylation positively modulates TCR signalling. The reliance on S-acylation was most apparent at the IS, the specialised membrane contact area that form at the interface between T cells and antigen presenting cells. Using SEE-pulsed Raji cells to induce IS formation, NFAT-dependent transcription was severely impaired in Jurkat T cells bearing the C143A mutant. We observed three major synapse assembly defects in Jurkat T cells reconstituted with ORAI1-C143A. First, ORAI1-C143A was less efficiently recruited to the IS and accumulated in the opposite pole, forming cap-like structures. ORAI1 enrichment in distal caps was previously reported and proposed to serve as a source of preformed channel complexes moving to new IS *Bascom, 1991*; *Barr et al., 2009*. Second, actin was less enriched in IS forming in cells reconstituted with ORAI1-C143A, indicative of a weakened interaction. The IS contains a high percentage of highly ordered lipids *Zech et al., 2009* forming lipid rafts migrating to its periphery *Owen et al., 2012*. S-acylation might target ORAI1 channels to these cholesterol-rich regions to optimise $Ca^{2+}$ signalling efficiency at the IS (*Figure 6H*). Third, TCR accumulation and signalling were reduced in cells reconstituted with the S-acylation-deficient channel. During strong antigenic stimuli, TCR form clusters with associated scaffolding and signalling proteins that segregate in three concentric zones of the IS *Monks et al., 1998*; *Grakoui et al., 1999*. Defective ORAI1 targeting might impact TCR dynamics in several ways. In quiescent T cells, $Ca^{2+}$ fluxes across ORAI1 channels might disrupt the CD3-lipid interactions that

prevent spontaneous TCR phosphorylation *Shi et al., 2013*, enhancing the activity state of TCR prior to their engagement. ORAI1 targeting to specialised PM domains such as filopodia might be required for this priming effect to occur. Preventing ORAI1 targeting could also impact the location or activity of integrin receptors such as ICAM-1, thereby indirectly altering TCR signalling. Further experiments are required to establish whether the defective IS re-localisation of C143A ORAI1 reflects the reduced levels of $Ca^{2+}$ entering across the deacylated channel, which might impair the movement of ER-PM junctions to the IS. It will be interesting to explore also if ORAI1 S-acylation promotes its endocytosis and whether dynamic S-acylation impacts the affinity of ORAI1 for accessory proteins or its potential interactions with other channels such as TRPC.

In summary, our findings establish that S-acylation by PAT20 (zDHHC20) at Cys143 regulates clustering and function of the ORAI1 channel. PAT20-mediated S-acylation targets ORAI1 to lipid-ordered PM domains rich in cholesterol and is required for $Ca^{2+}$ signals that drive Jurkat T cells proliferation and for efficient TCR signaling in immune synapses. We propose that S-acylation dynamically targets the ORAI1 channel to microdomains rich in cholesterol during immune synapse formation to ensure efficient Jurkat T cell signalling following TCR engagement.

## Materials and methods
### Antibodies and reagents
The following reagents were used in this manuscript; Thapsigargin (T9033/CAY10522, Sigma); Ionomycin (I9657, Sigma); Phorbol 12-myristate 13-acetate (PMA) (79346, Sigma); Fura2-AM, (F1201, Invitrogen); Fluo-8, AM (21082, AAT Bioquest); SiR-Actin (Far Red, Spirochrome) Cyclopiazonic acid from Penicillium cyclopium, (c1530, Sigma); Gadolinium (G7532, Sigma); Vybrant Alexa Fluor 555 Lipid Raft Labeling Kit cholera toxin subunit B (V34404, Thermo-Fisher); Lipophilic Tracer Sampler DiD (L7781, Thermo-Fisher); Hoechst 33,342 (H3570, Thermo-Fisher); Dynabeads Human T-Activator CD3/CD28 (11,161D, Thermo-Fisher), GFP-trap agarose (GTA-10. Chromotek), - hydroxylamine (55460, Sigma, used at 0.5 M), Zebra spin desalting columns (PIER89882, Pierce)), 3H- palmitic acid (ART0129-25, American radio labelled chemicals), NEM (04559, Sigma), protein G (17-0618-01, GEHealthcare), 5 kDa PEG (63187, Sigma). For protein detection either on western blot or immunofluorescence, we used; NFATc1 (clone 7A6, MABS409, Sigma), TCR alpha/beta-PE (12-9986-42, eBioscience), Anti-Cholera Toxin, B-Subunit (227040, Sigma), Myc-Tag (9B11) (2276, Cell-Signalling), gamma Tubulin (4D11) (MA1-850, Thermofisher), ANTI-FLAG M2 (F1804, Sigma), anti-ORAI1 (600–401-DG9, rockland immunochemicals Inc), anti-GFP (SAB4301138, Sigma), anti-mouse-HRP, and rabbit-HRP (1706516 and 172101, Bio-Rad (USA).

### Cell culture, cell lines, and DNA constructs
Human embryonic kidney (HEK-293T) and Human retinal pigment ephitilial-1 (RPE1) cells were obtained from ATCC (CRL-11268, Manassas, VA, USA) maintained in Dulbecco's modified Eagles medium (cat. no. 31966-021) supplemented with 10 % fetal bovine serum and 1 % penicillin/streptomycin and grown at 37 °C and 5 % $CO_2$. HeLa cells purchased from the European collection of cell culture (ECACC) were grown in MEM Gibco (41,090 in the same conditions). HEK-293 cells CRISPR triple knockout for ORAI1, 2 and 3 were a kind gift from Dr. Rajesh Bhardwaj, University of Bern. HEK-293T Stable cell lines expressing Cherry-Stim1 and hORAI1-WT or mutant C126A, C143A or C1267C143A were generated for this article. Briefly, we first infected with Cherry STIM1 p2K7 lentiviral vector, sorted, and then infected for the indicated mutants at a MOI of 2 and sorted for the same Cherry-STIM1 and ORAI1-YFP intensity. Indicated constructs were subcloned into a pWPT vector and co transfected with pCAG-VSVG/psPAX2 into HEK-293T cells to produce viral particles as described in *Salmon, 2013*. Briefly, indicated constructs were subcloned into a pWPT vector and co transfected with pCAG-VSVG/psPAX2 into HEK-293T cells to produce viral particles. After accumulation, ultracentrifugation and titration of the virus these were stored at –80 °C. Jurkat T clone E6 cells were purchased from ECACC and grown in RPMI 1640 (21875-034 Life Technologies) supplemented with 10 FCS and 1 % Pen/Strep. CRISPR Jurkat T cells were generated for this study by stably expressing with lentiviral particles pLX-311-Cas9 construct (Addgene 96924) and transiently transfecting with Amaxa Cell Line Nucleofector Kit T (Ref: VCA-1002, Lonza) two sets of sgRNAs (Hs.Cas9.ORAI1.1.AA Ref: 224748421/ Hs.Cas9.ORAI1.1.AB Ref: 224748422, IDT). Single clone sorting, genomic DNA

sequencing and western blot were used to validate ORAI1 KO cells. ORAI1 rescue on CRISPR Jurkat T ORAI1 KO cells was performed originally for this article by infecting at a MOI of 5 and FACS sorting for YFP fluorescence. To avoid clonal effects all cells used or generated in this study were pooled populations with the exception of HEK-TKO for ORAI1/2/3 or Jurkat CRISPR ORAI1 cells which were validated by either transient (HEK-293) or stable (Jurkat) expression of the native or mutated proteins. All cells sorted in this study were generated using a Beckman Coulter MoFlo Astrios integrated in PSL2 hood. All cell lines from this study were tested negative for mycoplasma contamination. Jurkat, HEK 293 and HeLa cells are listed as commonly misidentified cell lines maintained by the International Cell Line Authentication Committee. In our hands, HeLa and HEK 293T cells were genetically confirmed (by genomic profiling [STRs]) prior to stockage and Jurkat E6.1 cells were purchased from ECACC (ref number 880442803).

The ORAI1-yellow fluorescent protein (YFP) construct was purchased from Addgene (Cambridge, MA, USA; plasmid no. 19756). Site-directed mutagenesis using the Pfu Turbo DNA polymerase from Agilent Technologies ((Santa Clara, CA, USA; 600250) was used to introduce Cysteine mutants C143A, or S, and C126 A or s). Forward (fwd) and complementary reverse mutagenesis primers (Mycrosinth [Balgach, Switzerland]) were as follows: C143A fwd: 5'-GCG CTC ATG ATC AGC ACC gcC ATC CTG CCC AAC ATC GAG GC-3', C143S fwd: 5'-GCT CAT GAT CAG CAC CaG CAT CCT GCC CAA CAT CG-3', C126A fwd: 5'-GCT CAT CGC CTT CAG TGC Cgc CAC CAC AGT GCT GGT GGC-3', C126S fwd: 5'-GCT CAT CGC CTT CAG TGC CaG CAC CAC AGT GCT GGT GGC-3'. All plasmids encoding for human DHHC1-24 were Myc tagged in the N-terminus in pcDNA3 vectors, kindly provided by the Fukata lab. 9NFAT-and the Renilla luciferase [phRL-TK-luc, Promega] were a kind gift from Stephan Konig. Antibodies against total and phosphorylated Erk1/2 were a kind gift from Prof. Bernhard Wehrle-Haller.

## Radiolabelling 3H-palmitic acid incorporation

To follow S-acylation, transfected or non-transfected cells were incubated 1 hr in medium without serum (Glasgow minimal essential medium buffered with 10 mM Hepes, pH 7.4), followed by 2 hr at 37 °C in IM with 200 µCI /ml 3 H palmitic acid (9,10-3H(N)), washed with cold PBS prior immunoprecipitation overnight with anti-ORAI antibodies and protein G-beads or anti-GFP agarose-coupled beads. Beads were incubated 5 minutes at 90 °C in reducing sample buffer prior to SDS-PAGE. Immunoprecipitates were split into two, run on 4–20% gels and analysed either by autoradiography (3H-palmitate) after fixation (25 % isopropanol, 65 % H2O, 10 % acetic acid), gels were incubated 30 min in enhancer Amplify NAMP100, and dried; or western blotting.

## Acyl-Peg-exchange

To block free cysteine, cells were lysed and incubated in 400 µl buffer (2.5 % SDS, 100 mM HEPES, 1 mM EDTA, 40 mM NEM pH 7.5, and protease inhibitor cocktail) for 4 hr at 40 °C. To remove excess unreacted NEM, proteins were acetone precipitated and resuspended in buffer (100 mM HEPES, 1 mM EDTA, 1 % SDS, pH 7.5). Previously S-acylated cysteines were revealed by treatment with 250 mM hydroxylamine ($NH_2OH$) for 1 hr at 37 °C. Cell lysates were desalted using Zebra spin columns and incubated 1 hr at 37 °C with 2 mM 5 kDa PEG: methoxypolyethylene glycol maleimide. Reaction was stopped by incubation in SDS sample buffer for 5 min at 95 °C. Samples were separated by SDS-PAGE and analysed by immunoblotting.

## Ca²⁺ imaging and plate reader

Calcium assays in single-cell live imaging (HeLa, HEKS1/O1^WT, HEK S1/O1^C143A or HEK TKO) were performed as described previously *Nunes et al., 2012*. Briefly, cells were transfected when indicated and seeded on a Poly-L lysine-coated coverslip. The day after they were loaded with 3 µM Fura-2-AM, in modified Ringer's for 30 min at room temperature (RT). 340/380 nm excitation and 510 ± 40 nm emission ratiometric imaging was performed every 2 s. SOCE activity was triggered by emptying the ER stores by blocking SERCA with Thapsigargin in a Ca²⁺-free solution containing 1 mM EGTA instead of 2 mM CaCl₂. Extracellular calcium addition revealed ORAI1 activity. Jurkat cells attachment to the coverslip was achieved by seeding 200,000 cells in 25 mm poly-L lysine-coated coverslips for 25 minutes at RT. When indicated, Jurkat cells were transfected with YFP cameleon (YC 3.6) calcium cytosolic probe to measure cytosolic calcium. YC 3.6 was excited at 440 nm and emission was collected

alternatively at 480 and 535 nm. Fura-2 Jurkat experiments were performed at 37 °C. Calcium imaging in plate reader was achieved using Fura2 as calcium Dye. Fluorescence was measured using a 96-well microplate reader with automated fluid additions at 37 °C (FlexStation 3, Molecular Devices).

## TIRF imaging

TIRF imaging to determine ORAI and STIM clusters in HEK-293 cells S1/O1 was performed on a Nikon Eclipse Ti microscope equipped with a Perfect Focus System (PFS III) using a 100× oil CFI Apochromat TIRF Objective (NA 1.49; Nikon Instruments Europe B.V.). To observe STIM1/ORAI1 clusters cells were bathed with CPA 10 µM and imaged every 20 s in calcium-free solution. ORAI1-YFP was imaged using ZET488/10 excitation filter (Chroma Technology Corp.). STIM1 cherry clusters were imaged using a ZET 561/10 excitation filter (Chroma Technology Corp., Bellows Falls, VT). All emission signals were collected by a cooled EMCCD camera (iXon Ultra 897, Andor Technology Ltd). All experiments were performed at room temperature (22–25°C). Image analysis was performed with a GitHub script (https://github.com/Carandoom/STIM-ORAI-Segmentation; *Henry, 2021*). Segmentation was performed at 600 s for each channel and a mask encircling STIM1 puncta was generated from three consecutive STIM1 images to graph the average ORAI1 fluorescence within STIM1 puncta over time.

## Confocal live imaging; FRAP

Confocal time lapse microscopy was used to image Fluorescence recovery after photobleaching.

FRAP was performed in HEK-293 S1/O1 under resting condition using the same microscope. ORAI1 FRAP was accomplished by following the protocol previously described *Zhou et al., 2018*. Briefly, we used a live chamber at 37 °C and 5 % CO2. Pinhole was settled at 1AU and images were sampled every 3 s for 100 images. Bleaching was for 20 s (488 nm 100 % output) after 1 min of basal acquisition. ROI of interest was compared to the same size ROI in the same field of view and normalised to basal. Traces were fitted with an exponential one-phase association model to obtain the half-life, $\tau$ 1/2 and fluorescence recovery. Diffusion coefficient was calculated with the formula $D = 0.224r^2/(\tau\ 1/2)$, in which r is the radius of the bleached circle region as described in *Zhou et al., 2018*.

## Flow cytometry

Cytometry calcium experiments on Jurkat cells were performed by incubation with Fluo8 (2 µM 30 min, RT) and washed for 15 min in a calcium containing solution. BD Accuri C6 was used to measure calcium movements over time by setting the flow at 1 µL per second. Every experiment started with $5 \times 10^5$ cells in 510 µL of calcium-free solution (1 mM EGTA). After 1 min 50 µL of Tg 10 µM was added to empty ER stores. After 300 s, we added 100 µL of CaCl$_2$ (Final concentration 2.5 mM) to reveal SOCE. IL-2 measurements were performed as described previously. Briefly, Jurkat cells (50.000) were transfected (when indidcated before) seeding in pre-coated CD3 (OKT3) round bottom 96 cell plates for 1 days. Cells were then fixed (PFA 4%) and perm/blocked with PBS-2%BSA 0.5 % Saponin previous to IL-2 PE incubation. IL-2 FACS measurements were acquired in a BDLSR Fortessa unit.

## Giant plasma membrane vesicles (GPMV)

GPMVs were formed and analysed following this protocol *Sezgin et al., 2012*. Briefly, HEK-293 cells were seeded in poly-L-lysine-coated 25-mm glass coverslips and transfected with a PI(4,5)P2 lipid selective protein (PH domain of PLC-delta) fused to mCherry, empty vector or PAT20 and YFP-ORAI1 WT or the C143A mutant. The day after cells were washed with GPMV buffer (150 mM NaCl, 2 mM CaCl2 and 10 mM HEPES, pH: 7.4) and incubated for 1 hr at 37 °C 5 % CO2 with a vesiculation buffer (25 mM PFA, 2 mM DTT). Cell super natant was then spun for 5 min at 100 x g. Supernatant were incubated with Alexa-647 Cholera Toxin B subunit 10 minutes on ice. Imaging was performed at 10 °C using Open Perfusion Microincubator (PDMI-2, Medical Systems, Greenvale, NY) temperature controller to enhance lipid partitioning. Vesicles were imaged using a Nikon A1r Spectral with a 60 × 1.4 CFI Plan Apo Lambda WD:0.13 mm objective using 488, 551 and 639 laser lines. Raft preference was calculated as on *Sezgin et al., 2012*, briefly, line scan profiles of ORAI in a Cholera toxin B positive region compared to a PiP2 positive region within the same GPMV was used to calculate the % of ORAI that distributed to raft with the following formula

$$ORAI1\ Raft\ preference = \frac{ORAI1RAFT}{(ORAI1RAFT + ORAIPiP2)} * 100$$

## Electrophysiology

$I_{CRAC}$ currents were recorded using the whole-cell configuration in HEK-293 cells stably expressing mCh-STIM1 and ORAI1-YFP (O1/S1) bearing or not the C143A mutation. The cells were trypsinised, seeded on 35 mm dishes (Corning, NY, USA) and incubated overnight at 37 °C to allow attachment of separated cells. The experiments were performed at room temperature. Pipettes were pulled from 1.5 mm thin-wall glass capillaries (GC150TF, Harvard Apparatus) using a vertical PC–10 Narishige puller to obtain a resistance between 2 and 4 MΩ. Currents were recorded with pCLAMP 10.7 software (Molecular Devices, Sunnyvale, CA, USA), using the Axopatch 200B amplifier (Axon Instruments, Molecular Devices) with a low–pass filtering at 1 kHz, and digitised with the Axon Digidata 1,440 A at 1 ms. Voltage ramps of 180 ms were applied from –120 to +100 mV every 5 s from a holding potential of 0 mV. Peak current densities ($I_{max}$) were measured at –100 mV after subtraction of basal or 10 µM GdCl$_3$-insensitive currents. The standard 10 mM Ca$^{2+}$ recording solution contained 130 mM NaCl, 5 mM KCl, 1 mM MgCl$_2$, 10 mM CaCl$_2$ and 10 mM HEPES (300–310 mOsm, pH 7.4 adjusted with NaOH). The intracellular pipette solution contained 130 mM Cs methanesulfonate, 8 mM MgCl$_2$, 10 mM BAPTA, and 10 mM HEPES (290–300 mOsm, pH 7.2 adjusted with CsOH).

Fast Ca2+-dependent inactivation (FCDI) was recorded by applying step voltage pulses of 200 ms at −120,−100, −80, and −60 mV every 5 s from a holding potential of 0 mV after the current was fully developed. The current was gradually developed following slow ER calcium depletion by 10 mM EGTA contained pipette solution. The fraction of remaining current was obtained by dividing the current at 197 ms from the start of the pulse in each voltage step pulse by the current recorded at 3 ms. The time course of Orai1 current inactivation was fitted with the one-phase exponential function from 0 ms to 50 ms, $I = I_0 + A_1 e^{-t/\tau 1}$, where $I$ is current, $I_0$ is the current at 1 ms, $A_1$ is amplitude ($I_{50ms} - I_0$), and $\tau 1$ is inactivation time constants.

## NFAT-luciferase assay

A total of $2 \times 10^6$ Jurkat cells were electroporated with Amaxa nucleofector (Kit V) with 1 µg of firefly luciferase encoding plasmid 9NFAT-luc together *Konig et al., 2006* with 0.2 µg of control plasmid encoding the Renilla luciferase (phRL-TK-luc, Promega). After 24 hr, 80,000 cells per biological replicate were treated with the indicated compounds or co-cultured with RAJI and processed with the Dual-Luciferase reporter assay kit (Promega) as recommended by the manufacturer. Luminiscence was measured in a SpectraMax L 384 w (Molecular devices).

## IS live cell imaging

Jurkat IS formation with Raji cells was performed as follows: Raji cells were seeded in fibronectin coated coverslips for 1 hr in a medium containing SEE 1 µg/ml for and CellMask Deep Red Plasma membrane Stain (Thermo, C10046) at 1:5000 dilution. After cells were washed three times with RPMI medium and Jurkat cell were added at a ratio of 1:1. Jurkat cells were identified with a green channel (488 excitation) and RAJI with far red (620 excitation). Stacks of 3 µm covering all cells were taken for both channels every 30 s for 30 min. Experiments were performed at 37°C .

## Immunofluorescence

For NFAT translocation Jurkat cells were treated with Tg 1 µM for the indicated times and seeded into poly-L lysine coated coverslips for 15 min at RT. For CD3, coated coverslips were used for the indicated times. After, cells were fixed (PFA 4%) for 20 min at RT, then permeabilised (PBS-BSA 0.5 %+ NP-40 0.1%) for 10 min and then blocked (PBS-BSA 0.5 %+ FBS 5%) for 1 hr at RT. Then, cells were incubated with primary antibodies (NFATC1, MABS409, Sigma) at 4 °C then with secondary 1:1000 with Hoesch 1:5000 for 1 hr at RT. NFATC1 analysis was done by dividing the nuclear to the cytosolic (total/nuclear) pixel intensity per cell into three to five randomised fields per condition. Jurkat IS formation with RAJI cells was performed as follows: Raji cells were seeded in fibronectin coated coverlips for 1 hr in a medium containing SEE 1 µg/ml for and CellMask Deep Red Plasma membrane Stain (Thermo, C10046) at 1:5000 dilution. After cells were washed three times with RPMI medium and Jurkat cell were added at a ratio of 1:1. After 1-hr cells were fixed with PFA (4%) for 20 min and stained for TCR alpha/beta ((IP26)), eBioscience in PBS-BSA 2 % for 15 min at RT. We then stained with DyLight 350 Phalloidin in PBS + NP-40 0.1 % for 10 min at RT and mounted coverslips after a fast wash in distilled water. Images were obtained in a LSM700 Zeiss Axio Imager.M2 microscope.

IS accumulation over Opposite pole was calculated as previously described *Quintana et al., 2011*. Briefly, line scan profiles from IS to opposite poles of Jurkat T cells were used to obtain Peak intensities of either ORAI1, Actin or TCR fluorescences in both compartments. The distribution from IS to opposite pole for each protein was calculated with the following formula:

$$IS\ enrichment = \frac{IS\ fluorescence}{Opposite\ Cap\ fluorescnece}$$

## Cell surface protein isolation

Cell surface biotinylation was performed with the Pierce Cell Surface Isolation Kit (Thermo Scientific; 89881), according the manufacturer's protocol with minor adjustments. In summary, Jurkat cells were expanded to $6 \times 10^5$ cells/mL on the day of experiments and 50 mL of suspension cells were used. All centrifugation of cells were carried out at 800 x g for 5 min. Cells were washed with room temperature PBS once and incubated with EZ-LINK Sulfo-NHS-SS-biotin for 10 min at room temperature. Cells were then washed with cold TBS twice and lysed with 500 mL lysis buffer (from kit, with addition of protease inhibitor cocktail (Sigma Aldrich; S8820-2TAB)) on ice for 30 min. Protein lysate were pelleted at 15,000 x g for 5 min at 4 °C and 30 ml of supernatant (total) were collected for immunoblot analysis. The biotinylated cell surface proteins were isolated using NeutrAvidin agarose containing column, washed with wash buffer and eluted with 200 ml elution buffer with DTT. 4 x sample buffer (ThermoFisher; NP0007) with b-mercaptoethanol were added for immuoblotting analysis. Immunoblots were probed with antibodies against GFP for Orai1 detection, STIM1, GAPDH, and PMCA.

## Image analysis and statistics

All images were analysed using ImageJ software or Matlab.

## Acknowledgements

We are grateful to Cyril Castelbou for the technical assistance, the bioimaging, READS and flow cytometry facilities (Geneva Medical Centre). This work was funded by the Swiss National Foundation [grant number 310030_189042 (to ND) and SNF 310030B_176393 and 310030_192608] European Research Council under the European Union's Seventh Framework Programme (FP/2007–2013) / ERC Grant Agreement n. 340260 - PalmERa' (to GvG).

## Additional information

### Funding

| Funder | Grant reference number | Author |
|---|---|---|
| Swiss National Science Foundation | 310030_189042 | Nicolas Demaurex |
| Swiss National Science Foundation | 310030B_176393 | F Gisou van der Goot |
| Swiss National Science Foundation | 310030_192608 | F Gisou van der Goot |
| European Research Council | FP/2007-2013 340260 | F Gisou van der Goot |

The funders had no role in study design, data collection and interpretation, or the decision to submit the work for publication.

### Author contributions

Amado Carreras-Sureda, Conceptualization, Data curation, Formal analysis, Investigation, Supervision, Validation, Writing – original draft, Writing – review and editing; Laurence Abrami, Data curation, Formal analysis, Investigation, Validation, Writing – review and editing; Kim Ji-Hee, Data curation, Formal analysis, Investigation; Wen-An Wang, Data curation, Formal analysis, Validation, Writing – review and editing; Christopher Henry, Formal analysis, Software; Maud Frieden, Formal analysis,

Supervision, Validation, Writing – review and editing; Monica Didier, Conceptualization, Data curation, Formal analysis; F Gisou van der Goot, Conceptualization, Funding acquisition, Investigation, Supervision, Validation, Writing – review and editing; Nicolas Demaurex, Conceptualization, Funding acquisition, Investigation, Project administration, Supervision, Writing – original draft, Writing – review and editing

Author ORCIDs
Amado Carreras-Sureda [ID] http://orcid.org/0000-0002-9032-5639
Laurence Abrami [ID] http://orcid.org/0000-0002-1774-0481
Wen-An Wang [ID] http://orcid.org/0000-0003-3871-0174
Christopher Henry [ID] http://orcid.org/0000-0002-3243-882X
Maud Frieden [ID] http://orcid.org/0000-0001-7135-0874
F Gisou van der Goot [ID] http://orcid.org/0000-0002-8522-274X
Nicolas Demaurex [ID] http://orcid.org/0000-0002-9933-6772

Decision letter and Author response
Decision letter https://doi.org/10.7554/eLife.72051.sa1
Author response https://doi.org/10.7554/eLife.72051.sa2

## Additional files

### Supplementary files
• Transparent reporting form
• Source data 1. Excel sheet with all statistical analyses performed for every figure on this paper.
• Source data 2. Document with the full scans for Wester blots corresponding to the original paper figures. Red rectangles highlight the images used on the manuscript.

### Data availability
All data generated or analysed during this study are included in the manuscript and supporting files.

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
