## [Editor Report]

This study uses a wide range of approaches to identify acylation as a key regulator of Orai1 channel activation and calcium entry. The results show a novel role for Orai1 acylation in promoting signaling and early activation events upon interaction of T cells with antigen-presenting cells.

---

## [Decision Letter]

**Decision letter after peer review:**

[Editors’ note: the authors submitted for reconsideration following the decision after peer review. What follows is the decision letter after the first round of review.]

Thank you for submitting your work entitled "S-acylation targets ORAI1 channels to lipid rafts for efficient Ca^2+^ signaling by T cell receptors at the immune synapse" for consideration by *eLife*. Your article has been reviewed by 3 peer reviewers, one of whom is a member of our Board of Reviewing Editors, and the evaluation has been overseen by a Senior Editor. The following individual involved in review of your submission has agreed to reveal their identity: Murali Prakriya (Reviewer #3).

While all three reviewers expressed considerable interest in the work, they identified a significant number of essential revisions that will be needed to support the main conclusions. These will likely require more time than *eLife* policy allows for inviting revisions (approximately two months). Thus, we are sorry to say that we cannot further consider this manuscript for publication by *eLife*. However, if you choose to address these concerns, we would be willing to consider a revised manuscript as a new submission. To assist you, we list below the essential experiments that would be required:

1. Measure the expression of C143A Orai1 in the PM, to address the possibility that C143A reduces Orai1 activity through altered trafficking.

2. Compare the SOCE responses of WT and C143A Orai1 after knocking down the various PATs, to determine whether C143A reduces Orai1 activity through acylation-independent mechanisms.

3. Obtain clearer data to characterize Orai1 WT and C143A targeting to lipid domains.

4. Resolve the inconsistent effects of up- and downregulating PATs to clarify which PATs are responsible for Orai1 S-acylation.

5. Obtain more consistent data relating Ca signals, NFAT translocation and IL-2 secretion in Jurkat cells.

6. Perform the immune synapse experiments under conditions that will allow conclusions about physiological effects of acylation on immune cell activation. While conducting studies with primary T cells would be preferable, the reviewers agreed that Jurkat cells could be used as a first model here, but only at a physiological temperature that fully supports synapse formation and with a more physiological substrate (e.g., supported lipid bilayers) that would allow movement of TCR complexes in the plasma membrane. These experiments would likely be the most time-consuming, but are needed to support the impact of S-acylation on immune function, a main area of significance for the paper.

*Reviewer #1:*

This paper reports that the store-operated calcium channel Orai1 is S-acylated under physiological conditions, and that acylation enhances its activity, promoting calcium signaling, transcriptional activation, and immunological synapse function in activated T cells. The authors present strong evidence that Orai1 palmitoylation at cysteine 143 is carried out by several acyltransferases, most significantly ZDHHC20 (PAT20). A particular strength of the paper is the use of stable Jurkat T cell lines in which endogenous Orai1 was replaced with matched levels of wild type or acylation-resistant mutant (C143A) Orai1, in order to study the effects of acylation on calcium signals and downstream effects. C143A Orai1 has reduced activity in response to passive ER calcium depletion, as well as a slower activation rate and diminished inactivation over time. Importantly, the authors examine the physiological impact of acylation in Jurkat T cells expressing WT or C143A Orai1 upon stimulation through the T cell antigen receptor (TCR). C143A Orai1 diminishes calcium signals, NFAT activation and IL-2 generation, as well as several aspects of immune synapse formation, including localization of Orai1, TCR clusters, and actin rings to the synapse after TCR stimulation, suggesting that acylation of Orai1 is required to support efficient immunological function.

There are several weaknesses in the paper that will need to be addressed in order to support the major conclusions.

1) The conclusion that PAT20 is responsible for acylation in vivo needs stronger support. Several acyltransferases are shown to palmitoylate Orai1, apparently differing in their preference for long or short isoforms of Orai1, and there are discrepancies between the effects of knocking down vs. overexpressing PAT isoforms on store-operated calcium entry (SOCE) amplitude. Potential effects of the C143A mutation on channel function independent of acylation have not been addressed.

2) Mechanistically, it is unclear how acylation affects Orai1 activity. Although C143A Orai1 was shown to form smaller clusters with STIM1 upon activation, both WT and mutant Orai1 clustered with abnormally slow kinetics relative to published reports, lessening confidence in this finding. Altered partitioning of the mutant Orai1 into ordered and disordered lipid domains needs more thorough demonstration, and how it could explain reduced channel activity is not clear. Finally, it is not clear whether the altered signaling properties of C143A arise from channel itself or from aberrant trafficking to the cell surface.

3) The conclusion that acylation is required for efficient activation of T cells is undermined by weaknesses in the supporting data. The quantitative relations between calcium signal amplitude, NFAT activation and IL-2 production are inconsistent. Experiments on immune synapse formation in vitro (including localization of Orai1, TCR clusters, and actin ring at the synapse) were conducted at low temperatures that do not support efficient synapse formation, making it difficult to estimate the true effects of acylation on these aspects of T cell activation under more physiological conditions.

Essential revisions:

1. The evidence for acylation at C143 is clear, but I am not convinced a single enzyme (PAT20) is responsible for it.

a) Overexpression of a number of PAT proteins can increase 3H-palmitate incorporation in two bands which do not migrate at the same MW as Orai (Figure 4A). Either this is an error in labeling the gel, or another protein that co-ip's with Orai1 is the target.

b) There is no evidence that the doublet of bands corresponds to Orai1alpha and Orai1beta, and the authors' claim that PAT20 only acylates the high MW band while PAT3 and PAT7 acylate the lower MW band is not strictly true. All three acylate both forms, with a slight preference for one over the other. To make this conclusion, the authors would need to express the α or β forms of Orai1 separately and show differential effects on acylation.

c) It seems inconsistent that knockdown of all 3 PATs (3, 7, and 20) reduce SOCE equally (Figure S4B), yet only overexpression of PAT20 enhances SOCE (Figure 4), particularly since PAT3 and PAT7 were not knocked down to nearly the same extent as PAT20 (Figure S4B) and PAT20 was hardly overexpressed (Figure 4B).

d) The partial overlap of Orai1 and PAT20 fluorescence in the cell does not add significant support to the role of PAT20 in acylating Orai1 (Figure 4D).

2. The reduction of Orai1 activity by the C143A mutation is clearly demonstrated, but the effects on activation kinetics and inactivation are not. The activation time course could be simply measured by time to half peak or 90% peak current; I am not sure what shifting the traces in time to align them by their "inflection points" signifies physiologically. With regard to inactivation, are the authors referring to Ca^2+^-dependent inactivation, and if so, how would this occur in the presence of 10 mM BAPTA in the pipette? A mechanistic description of this is not necessary, but more examples of the inactivating currents should be shown in a supplementary figure, with an average time course to characterize the overall effect.

3. Does C143A alter Orai1 activity only by preventing acylation, or could it have acylation-independent effects? If it is only due to prevention of acylation, then eliminating PAT activity should have the same effect in cells expressing WT Orai1 (Figure S4B) and Orai1 C143A. This comparison should be made.

4. To pursue a mechanism to explain how C143A reduces Orai1 activity, the authors describe effects on diffusion rate, Orai1 puncta size, and loss of partitioning into cholesterol-rich domains. However, these data do not provide clear support for a unifying mechanism.

a) Diffusion rate. The FRAP experiments are nicely done, but the slowing of diffusion is slight (~15% in Figure 3C). It is not clear how this would lead to a large change in coupling to STIM1 as hypothesized in the Discussion (lines 236-238).

b) Orai1 puncta size. The time course of Orai1 cluster size (Figure S3) is puzzling and inconsistent with published data. Even with WT Orai1, size continues to increase slowly well after STIM1 has reached a plateau. In previous reports on HEK cells, Orai1 accumulates only slightly (~10 s) after STIM1 (e.g., Figure S3 of Perni et al., PNAS 112:E5533, 2015). It does not make sense that Orai1 cluster size would continue to increase for a long time, when STIM1 size is constant, considering that Orai1 clusters by binding to STIM1. Perhaps this is a problem with the method used to analyze Orai1 puncta fluorescence? A more direct and interpretable approach would be to measure by TIRF the level of Orai1 (and STIM1) fluorescence in the cell footprint over time. Does C143A reach a lower plateau fluorescence, hence a reduced number of channels able to conduct Ca^2+^? Perhaps this could be explained by diminished coupling to STIM1.

c) Lipid partitioning. I have concerns about the partitioning of Orai1 WT and C143A in raft and non-raft domains based on the quality of the images in Figure 3D. Why is the distribution of DiD and CxB so different in the upper and lower rows? C143A seems to have a non-uniform distribution that does not correspond to either the CxB or DiD distributions. The method of analysis is not described, and the different categories in the bar plot in Figure 3D are not explained. The authors should supply control data to show that the method yields meaningful results.

5. To explain the differences in Ca^2+^ signals in Jurkat cells expressing WT or C143A Orai1, it is imperative to know whether they arise from altered expression at the cell surface vs. altered properties of the channels themselves. Acylation has been shown to affect many aspects of ion channel biosynthesis, maturation, trafficking and localization as well as function. FACS analysis of cells stained with anti-Orai1 antibody can be used to measure surface expression (with appropriate controls). and a comparison of surface labeling with total GFP fluorescence (e.g. Figure S5D) can then be used to determine whether acylation affects trafficking to or from the plasma membrane. Were all copies of the endogenous Orai1 gene eliminated in the CRISPR Jurkat cells?

6. The authors conclude that acylation is required for efficient NFAT translocation to the nucleus, but the supporting data are inconsistent. With TG, KO of Orai1 causes only <50% decrease in NFAT translocation (Figure 5D), whereas a nearly complete suppression would be expected based on the SOCE response, and C143A almost completely rescues SOCE (Figure 5A), yet does not rescue NFAT translocation at all relative to the KO (Figure 5D). In response to anti-CD3, C143A does not promote detectable NFAT translocation, yet it stimulates IL-2 production by ~50% relative to WT Orai1 (Figure 5F). Were these experiments done at the same temperature? The conditions are not adequately described in the Methods. The lack of internal consistency of these results lessens confidence in the conclusion that Orai1 acylation is required for T cell transcription by TCR engagement.

7. The authors conclude that C143A Orai1 interferes with actin accumulation, Orai1 localization, and TCR cluster formation at the immune synapse. There are several problems with the data and experimental design that undermine this conclusion.

a) The proper formation of an artificial synapse on anti-CD3 coated coverslips is highly temperature dependent and does not occur efficiently at room temperature, the condition under which these experiments were done. For this reason, IS studies are always conducted at 37 deg C (see papers by Bunnell, Hammer, Dustin groups). Because low temperature may exacerbate effects of reduced Ca^2+^ entry on the synapse, it is difficult to use these results to infer how much acylation might affect synapse formation under physiological conditions.

b) The quality of the actin images is low in comparison to published work, making it difficult to judge the degree to which its structure has been changed by C143A Orai1. Given the quality of the actin images, it is hard to tell how actin rings and center, middle, and peripheral regions (cSMAC, pSMCA, dSMAC?) were detected and quantitated (Figure S6). Generally, actin filament organization in living cells is described using labeled probes like LifeAct or F-tractin (e.g., Yi et al., Mol. Biol. Cell 23:834, 2012). One of these probes should be tried, as SiR-actin could interfere with normal actin dynamics due to its structural similarity to jasplakinolide.

c) How do TCRs rearrange to accumulate in the center of the contact zone when they are bound to surface-attached, immobile anti-CD3, and in the movies why do the PE-labeled anti-TCR antibodies appear to move (the time scale should be stated)? Are we looking at internalized TCR?

*Reviewer #2:*

In previous studies, Orai1 was identified as a highly S-acylated protein in human T cells, as well as other murine neural stem cells. The main finding of this study is that S-acylation of Orai1 on Cys143 by PAT20 (or ZDHHC20) regulates CRAC function, lipid mobility and ultimately cytokine production by T cells upon TCR stimulation. Using acyl-PEG exchange, 3H-palmitate incorporation and mutagenesis of cystein residues, the authors show that Orai1 is S-acylated on Cys143, a highly conserved Cys residue, in HeLa cells. Authors show that PAT20, expressed in T cells, can S-acylate Orai1 and modulate SOCE. The authors then investigated the impact of Cys143 S-acylation on CRAC channel function by calcium imaging and patch clamping in HEK-293 and Jurkat cells and show a strong reduction in SOCE and ICRAC in cells expressing a C143A mutant (and thus S-acylation defective) form of Orai1. Furthermore, Orai1-C143A shows reduced mobility in membrane lipids, altering its accumulation in lipid rafts and efficient Orai1 clustering upon CRAC activation. Finally, authors demonstrate that S-acylation of Orai1 is required for robust calcium responses upon TCR stimulation in Jurkat cells leading to NFATc1 translocation and IL-2 production.

In Figure 3C, the effects of the Orai1-C143A mutant on lateral diffusion are very small. While statistically significant, it is unclear whether this difference is really meaningful. It would be useful if the authors could show the primary data for mutant Orai1 in addition to Orai1 WT.

The conclusion from Figure 4 that PAT20 S-acylates Orai1 is plausible, but the argument could be strengthened by performing palmitate incorporation in cells expressing Orai1 WT or Orai1-C143A together with PAT isoforms. This is shown only indirectly through measurements of SOCE in panel C.

Regarding the effects of PAT20 on SOCE shown in Figure S5F,G, I have a similar comment as in Figure 4: repeating this experiment with Orai1-WT and Orai1-C143A overexpression would strengthen the authors conclusions that PAT20 directly regulates Orai1 to modulate SOCE and IL-2 production instead of indirectly through acylation of another protein.

Figures 6A,B: From the representative images and quantifications shown I am not at all convinced that Orai1-WT localizes to the bead contact are (which the authors call immune synapse). If that were the case, why does the percentage of ORAI1-GFP at cups not increase over time (from 0 to 8 min)? The gray trace in Figure 6B is flat! The apparent difference between WT and C143A is only seen at one time point (8 min after stimulation) and results from reduced C143A at the cups. The quantification in Figure 6B is based on 5 and 3 cells expressing Orai1-WT and Orai1-mutant, respectively, which is not sufficient to draw these conclusions. Since the role of Orai1 acylation in IS formation and signaling features prominently in the title and abstract of the paper, these conclusions need to be solidified. (On a side note, looking at the data in the referenced papers (e.g. Ref. 21, Lioudyno et al), they are not that convincing either to support a recruitment of Orai1 to immune synapses in my opinion).

The most surprising finding of this study is that Orai1-C143A lacking acylation and promoting reduced SOCE impairs TCR clustering. This argument is entirely based on the anti-CD3 coating condition, and may as well be an artifact. Assuming Figure 6D-F show the quantification of data in 6C, it would be misleading to label the bar graphs as "Orai1-GFP at IS". Since bound anti-CD3 antibodies do not diffuse (as for instance GPI anchored MHC molecules in lipid bilayers), a proper IS cannot form in T cells. I would suggest to confirm this finding using another system to measure immune synapses, either the CD3 bead system used in 6A-B or even better lipid bilayers. The authors discuss a number of explanations in the Discussion of their paper, but these explanations or the hyposthesis itself that Orai1/SOCE regulates TCR clustering are not represented in the graphical abstract in 6G.

To make conclusions about the role of Orai1 acylation in T cell function, I would suggest to not only rely on a lymphoma cell line (Jurkat) but to confirm at least some of the key findings in real T cells. Alternatively, it would be appropriate to change the title and abstract of the paper.

Figure S4B: how do the authors explain that PAT20 (but not PAT3 or PAT7) overexpression increases SOCE, whereas siRNAs targeting PAT20, PAT7 or PAT3 all decrease SOCE to a similar extent? Are siRNAs specific to these isoform?

Figure 5 and S5: I am not quite convinced of the NFAT localization data. Are the graphs in Figure 5D the quantification of data in Figure S5E? How long were cells in 5D stimulated with thapsigargin? What does "NT" mean in S5E? In the representative data (?) in S5E, I do not see a difference in nuclear localization of NFATC1 in WT and mutant Orai1 transfected cells. How do the authors explain the discrepancy between Figure 5D and E with regard to (a) the NFATC1 nuclear /cytoplasmic ratio (approx. 6 in Orai1-WT cells after TG in D, but approx. 2 in Orai1-WT cells after CD3) and (b) the only small defect in D but complete defect in E?

*Reviewer #3:*

In this study, Carreras-Sureda and colleagues address the role of protein S-acylation in regulating the function of Orai1 channels. Orai1 contains three endogenous cysteine residues. Carreras-Sureda and co-workers find that the most intracellular of these, C143, located at the base of TM2, can be S-acylated. S-acylation was assessed using a straightforward PEG-5K and tritiated palmitate incorporation assays the test showed that a fraction of the cellular Orai1 pool is S-acylated. Mutational analysis traced the S-acylation site to C143. Measurements of SOCE and CRAC currents revealed that S-acylation of C143 significantly enhanced Orai1 activity, and mutating away C143 (to C143A) significantly diminished the amplitude of the STIM-gated Orai1 current. This is significant as reversible S-acylation of C143 could play an important role in modulation of Orai1 channel activity and hence Ca^2+^ signaling. Further, mechanistic analysis showed that blocking S-acylation (in the C143A mutant) decreases Orai1 mobility in the PM and decreases the size of the Orai1 punctae following store depletion, while STIM1 activation is unaffected. Moreover, the reduced mobility of Orai1 appears to result in exclusion of Orai1 from cholesterol-rich lipid rafts, raising the possibility that Orai1 fails to get into zones of signaling complexes when S-acylation is blocked. The authors also show using RNA suppression and protein over-expression that the protein acyltransferase enzyme, PAT20, is likely involved in S-acylating Orai1. Finally, using Jurkat T cells as a system, the authors examine the physiological role of Orai1 protein S-acylation. These studies indicated that stable transfection of C143A, but not WT Orai1, into Orai1-deficient Jurkat T cells results in impaired Ca signals following T cell receptor stimulation and this is accompanied by impaired T cell receptor assembly and signaling at the antigen-induced immune synapse.

Strengths: the discovery that Orai1 is S-acylated and this modification changes Orai1 function by impairing Orai1 mobility and formation of STIM1-Orai1 complexes is a significant discovery. It opens up the possibility that Orai1 S-acylation may represent a major mechanism for regulating SOCE and SOCE-dependent effector responses. This aspect of the study is compelling and nicely shown in the paper.

Weaknesses: Some claims of the underlying mechanism are not convincingly supported by the data shown and may have alternate explanations. Specifically, the experimental evidence for the conclusion that C143A does not affect Orai1 plasma membrane expression needs to be redone with better methods and more compelling results. Additionally, too many of the results seem to rely purely on the C143A mutant which may yet have un-anticipated effects. Therefore, additional tests using blockers of S-acylation would need to be added to support the main findings.

1) The assessment of Orai1 expression in the plasma membrane is an important variable and rigorous tests need to be done to check whether this key parameter is affected. In Figure S1C, the authors show widefield images of mCh-STIM1 and wildtype Orai1-GFP. The images seem to be low quality and it is difficult to assess whether mCh-Orai1 is significantly affected in the C143A mutant. What exactly was quantified in the summary bar graph next to the image? Plasma membrane expression? Total cell expression? How were the ROIs drawn? As a moderate drop in Orai1 expression could easily explain the reduction in SOCE and CRAC currents, this key parameter (Orai1 plasma membrane expression) needs to be determined with much greater precision to rule it out. Importantly, the quantification should be done using confocal microscopy and backed by additional confirmatory tests such as biotinylation of membrane proteins to determine whether plasma membrane Orai1 fraction is affected.

2) STIM1-Orai1 binding. It is unclear whether C143A affects STIM1 binding. One could envision that impaired Orai1 accumulation at the ER-PM junctions and ensuing changes in SOCE could result from impaired binding to STIM1. This should be assessed using STIM1-Orai1 FRET and/or other methods.

3) It would be interesting to determine whether C143A affects fast Ca^2+^ -dependent inactivation. A lack of affect would provide further support for the idea that the main change is the number of active Orai1 channels activated rather than a change in Orai1 channel properties.

4) Figure 3C: The diffusion rate differences between WT Orai1 and C143A Orai1 seem rather small. I am concerned that the difference is mainly driven by the one outlier data point in the WT group that shows very high D (~30 µm2/s). Please remove this outlier and run the test again to determine whether the data is significant. It is difficult to see how this relatively modest difference accounts for the very large difference in Figure S3A between WT and C143A Orai1 clusters at the late time points.

5) Related to the above points, I am also unclear on why HEK cells expressing both STIM1 and Orai1 were used for these experiments. If the goal is to assess the diffusion of Orai1 uncontaminated with binding to other proteins (such as STIM1), isn't the experiment best done in cells expressing only Orai1?

6) Figure 3D. The assessment of Orai1 distribution in rafts vs non-raft appears to have been carried out purely from widefield images. It is really difficult to see how this mode of imaging provides clear results. I would strongly recommend redoing these tests with confocal microscopy to improve the image quality and hence the robustness of the data.

7) Figure 4A should be quantified using densitometry analysis. In this blot, the levels of tritiated palmitate seem substantially lower (and barely present!) than what is shown in Figure 1. Why is that?

[Editors’ note: further revisions were suggested prior to acceptance, as described below.]

Thank you for submitting your article "S-acylation by zDHHC20 targets ORAI1 channels to lipid rafts for efficient Ca^2+^ signaling by Jurkat T cell receptors at the immune synapse" for consideration by *eLife*. Your article has been reviewed by 3 peer reviewers, one of whom is a member of our Board of Reviewing Editors, and the evaluation has been overseen by Kenton Swartz as the Senior Editor. The reviewers have opted to remain anonymous.

All three reviewers found the paper interesting and considered the most important findings to be the demonstration of Orai1 acylation at C143A and the resultant effects on Orai1 channel activation and NFAT/IL-2 activation in T cells after TCR stimulation. However, in order to provide mechanistic insight, extensive revisions will be needed along with new data to show that the effects of C143A are due to lack of acylation rather than direct effects on channel gating.

Essential revisions:

1. Many conclusions in the abstract are either overstated or not shown in the paper. These need to be revised or deleted.

a. "acylation of Orai1 is required for TCR assembly" (see also line 51). TCR assembly was not studied in this paper.

b. "The acylation-deficient channel accumulated in cholesterol-poor domains and failed to reach the IS, preventing proper IS formation." The data show no significant effect of C143A on accumulation in cholesterol-poor domains (Figure 3D), and C143A reaches the IS, just not as well as WT (Figure 6D, E). Also, the synapses were not characterized in sufficient detail to make a statement about whether they are "proper."

c. "S-acylation is a critical regulator of Orai1 channel assembly." Orai1 channel assembly was not studied in this paper.

d. "local Ca^2+^ fluxes are required for TCR recruitment to the synapse." Local Ca^2+^ fluxes were not measured, and TCR recruitment to the IS was not significantly altered with C143A (Figure 6G).

2. Because so many conclusions about the effects of acylation are based on C143A, it is essential to show that the behavior of C143A derives from the absence of acylation rather than a direct effect of the mutation on channel function. This could be done by comparing SOCE in C143A to WT+siPAT20; if effects are only due to inhibiting acylation, the two should be similar. The experiments of Figure S4B and C would allow such a comparison if the same [Ca^2+^] were applied in both. Likewise, to rule out the possibility that the C143A mutant has downstream effects unrelated to Orai1 s-acylation, NFAT activation and immune cell function should also be tested using PAT20 knockdowns in addition to the C143A Orai1 analysis.

3. The results show reduced accumulation of Orai1 in puncta, but analysis in terms of the size and number of Orai1 clusters makes it difficult to interpret. Cluster size and number is determined by the size and number of ER-plasma membrane junctions, which should not be affected by mutations in Orai1. In fact, STIM1 accumulation appears to be normal in C143A cells (Figure S3C). These results are probably due to a thresholding artifact; as intensity declines, the thresholded size and numbers of ROIs will also decline. The authors should instead measure the level of Orai1 accumulation in STIM1-containing puncta, which could be done by making a mask based on STIM1 puncta and applying it to Orai1 to get the average Orai1 fluorescence in puncta. This will indicate more directly the relative number of channels recruited to junctions.

4. It is not clear why only PAT20 overexpression enhances SOCE, when all three PATs (3/7/20) palmitoylate Orai1. The explanation given in line 228 about cycles of acylation and PAT20 being rate- limiting is not well developed. Why does only PAT20 enhance SOCE (Figure 4C) when all three proteins produce similar steady- state levels of palmitoylation (Figure S5E,F)?

5. The C143A current develops more slowly than WT, but this is likely because the currents and [Ca^2+^]i increase produced by C143A are much smaller than WT and elicit much less slow Ca^2+^-dependent inactivation. The large WT currents reach a peak earlier because they begin to inactivate, and this does not seem to occur with C143A. This point should be discussed.

6. The conclusion that C143A exhibits altered fast Ca^2+^ -dependent inactivation is intriguing but unconvincing. The initial phase of CDI in fact appears to be normal. What is different is the presence of secondary potentiation in C143A that is not seen in WT channels. Because of the way CDI is quantified (Iss/Ipeak), this metric would naturally be affected in the mutant. A previous study that looked at a cys-less Orai1 construct lacking all three endogenous cysteines concluded that CDI is qualitatively similar to WT Orai1 (PMID: 20018736), which would be consistent with the presence of inactivation that is also seen here. These results and conclusion should be restated to indicate that the mutant shows delayed time-dependent potentiation which clips off CDI that would otherwise have normally occurred (and not that CDI is impaired). [this is also consistent with the faster tau_fast in the mutant, Figure S2C]. More generally, the effects on CDI could arise from a direct effect of mutation or the lack of acylation on the CDI mechanism, or reduced STIM1 binding. Reduced affinity for STIM1 would be consistent with lower SOCE/ICRAC as well as reduced CDI. The authors should discuss these possibilities.

7. While C143A appears to diffuse a little more slowly than WT, it is hard to evaluate this effect because the D of WT Orai1 is much lower than reported by others (0.01 in this paper vs. ~0.08 μm2/s in ref 35 and Park et al., PMID 19249086). The authors should validate their method using a membrane protein with known diffusion coefficient; however, given the small magnitude of the effect, and uncertainty as to how it relates to the reduced activity of C143A, these data could be omitted.

8. It appears that the preference of Orai1 for rafts at 10 degC is not significantly affected by the C143A mutation, and the images do not show a noticeable difference either (Figure 3D). Both seem to be mostly excluded from CxB-labeled domains. The conclusion on line 146 is written as if Orai1 accumulation in rafts is somehow responsible for enhancing SOCE, but this seems unlikely given that C143A associates with raft lipids similarly to WT. Moreover, PAT20 has to be overexpressed to see a change in preference (Figure 4D), and yet the mean raft preference for WT+PAT20 in Figure 4D is the same as WT in Figure 3D, suggesting the measurements are not very reliable. Overall, given the small size of the C143A effect on partitioning, and the absence of a connection between raft localization and channel function, the relevance of these findings is limited and they could be omitted.

9. Figure 5F and G shows anti-CD3 fails to induce NFAT translocation with C143A Orai1, yet a substantial fraction of cells become IL-2 positive (2/3 of WT level). These results seem inconsistent given what we know about the role of NFAT in driving IL-2 expression.

10. The conclusion (line 194-5) that "These data indicate that ORAI1 S-acylation is required to recruit the ORAI1 channel and TCR to form a signalling-competent immune synapse in Jurkat T cells" is not supported by the data. The effects of C143A on actin recruitment (<10%) and TCR recruitment (<5%) are extremely small, which would be more apparent if the data were plotted from 0 instead of 50 (Figure 6G). How are the Orai1 localization results in 6D/E and 6G related? While the effects of C143A on SOCE, NFAT activation and IL-2 production are clearly significant, effects on the IS (TCR and actin accumulation) are not. Accordingly, the immune synapse data in Figure 6F-H and related parts of Figure S6 do not add significantly to the paper and could be deleted.

11. Line 240: "Reduced ORAI1 trapping by STIM1 would reduce CRAC currents and possibly FCDI by increasing the amount of STIM1 molecules required to trap and gate deacylated channels". STIM1 binding to Orai1 C143A could be measured using FRET. This test should be considered as the CDI results suggest that C143A may affect gating, perhaps due to decreased STIM1 binding.

12. Replicates and quantitation of the ERK phosphorylation will be needed to conclude that ERK signaling is inhibited by C143A. Only a single gel is shown.

*Reviewer #1:*

This paper reports that the store-operated calcium channel Orai1 is S-acylated under physiological conditions, and that acylation enhances its activity, promoting calcium signaling, transcriptional activation, and immunological synapse function in activated T cells. The authors present strong evidence in several cell lines that Orai1 is palmitoylated at cysteine 143 by several acyltransferases, most significantly ZDHHC20 (PAT20). C143A mutants cannot be acylated; cells expressing this mutant have a greatly reduced level of store-operated calcium entry (SOCE), as well as a reduced sensitivity to calcium-dependent inactivation. Similarly, knockdown of PAT20 reduced SOCE, while overexpression increased SOCE. These results strongly support the conclusion that acylation at C143 enhance the activity of Orai1 channels.

A particular strength of the paper is the use of stable Jurkat T cell lines in which endogenous Orai1 was replaced with matched levels of wild type or acylation-resistant mutant (C143A) Orai1, in order to study the effects of acylation on calcium signals and downstream events during T cell activation. C143A Orai1 significantly diminished calcium signals, NFAT activation and IL-2 generation evoked through the T cell receptor, either by stimulatory antibody-coated beads or contact with antigenic target cells, and impeded the localization of Orai1 to the immune synapse. The influence of acylation on specific aspects of immune synapse formation, including localization of actin and the TCR to the immunological synapse was less clear.

Mechanistically, it is unclear how acylation acts to enhance Orai1 activity. Altered partitioning of the mutant Orai1 into ordered and disordered lipid domains is possible, but how this could explain reduced channel activity is not understood. Also, it will be important to show that the effects of the C143 mutations on Orai1 and T cell function are due to altered acylation rather than a direct effect on channel activation.

*Reviewer #2:*

Orai1 contains three endogenous cysteine residues whose role in regulating channel function have been drawn much attention over the past decade. Carreras-Sureda and coworkers find that one cysteine, C143, located in transmembrane 2 can by modified via s-acylation, which impacts the activity and physiological role of Orai1 channels in T cells. PEG-5K and tritiated palmitate incorporation assays show that Orai1 is s-acylated and the site of acylation is narrowed to C143 from mutational analysis of the endogenous cysteines. The C143A mutation which should eliminate s-acylation, drastically reduced SOCE and CRAC currents independently of surface protein expression, suggesting that acylation is important for Orai1 gating and activity. Using the C143A mutant, the study shows that C143A Orai1 mobility in the plasma membrane is reduced and accumulation into STIM1 punctae following store depletion is impaired. Moreover, the mutant Orai1 appears to be excluded from cholesterol-rich lipid rafts, raising the possibility that Orai1 fails to get into zones of signaling complexes when S-acylation is blocked. Using RNA suppression and protein over-expression methods, authors also show that the protein acyltransferase enzyme, PAT20, is likely involved in S-acylating Orai1. Finally, using Jurkat T cells in which the endogenous Orai1 is CRISPered out and C143A Orai1 is expressed, the authors determine that stable transfection of C143A, but not WT Orai1, into Orai1-deficient Jurkat T cells impairs Ca signals following T cell receptor stimulation, T cell receptor assembly, NFAT signaling, and immune-synapse formation establishing the physiological relevance of s-acylation. Overall, the studies are well-done and the discovery that Orai1 is acylated is significant with potential to open up new avenues of research on Orai1 channel regulation. There are some key weaknesses which should be addressed to strengthen the conclusions.

[Editors’ note: further revisions were suggested prior to acceptance, as described below.]

Thank you for resubmitting your work entitled "S-acylation by zDHHC20 targets ORAI1 channels to lipid rafts for efficient Ca^2+^ signaling by Jurkat T cell receptors at the immune synapse" for further consideration by *eLife*. Your revised article has been evaluated by Kenton Swartz (Senior Editor) and a Reviewing Editor.

The manuscript has been improved but there are some remaining issues that need to be addressed, as outlined below:

(Comments are numbered according to the authors' rebuttal.)

1a. "Assembly" has been replaced by "recruitment," which is good. But the conclusion that "acylation of Orai1 is required for recruitment and signaling at the IS" is still an overstatement. TCR is actually recruited to the IS in C143A cells, as it is enriched by a factor of ~8, about half of WT (Figure 6G). Acylation enhances TCR recruitment and signaling at the IS but is not absolutely required for either. Please correct the same conclusion on Line 54-5.

1d. See response to 1a – acylation is not required for TCR recruitment to the synapse, rather it enhances or contributes significantly to it. The text gives the impression that without acylation there is no enrichment of the TCR at the IS, which is not true.

3. The new analysis of Orai1 fluorescence within STIM1 regions nicely addresses the comment. However, the STIM Orai1 segmentation video (from the github site) shows that the thresholding function does actually diminish the size of dim puncta relative to bright puncta. This will reduce the apparent size of the C143A Orai puncta since they are dimmer than the WT.

7. The authors may keep the data in the supplemental figure, as it does seem in agreement with one previous study (Zhou et al. 2018) and no claims are made about its contribution to reduced SOCE with C143A Orai1. However, I would like to make a comment. The authors propose that the very low diffusion coefficients measured here relative to some previous values are explained by differences in the levels of expressed proteins and the geometry of the bleached regions. However, it should be noted that single particle Orai1 measurements at low expression level (Wu et al., 2014) correspond well to FRAP measurements at high expression measured with rectangular bleach (Park et al., 2009). And for STIM1, a small circular bleach area (Liou et al., 2007) gave similar values of D as a rectangular bleach (Covington et al., 2010). More importantly, the slightly slowed diffusion of C143A Orai1, while it may slightly slow down the accumulation of Orai1 in puncta, is unlikely to affect the amount of Orai1 in puncta at steady-state, which is the determinant of activity. Thus, it is not clear how the reduced diffusion rate would contribute significantly to the loss of SOCE in the mutant.

10. Replotting the data in Figure 6G makes it easier to see the effect of C143A Orai1. It shows a reduction of Orai1, actin, and TCR accumulation at the synapse. However, it is important to note that these are all partial effects: all three are still enhanced at the synapse relative to the distal pole (Orai by 2-fold, actin by 3-fold, and TCR by 8-fold). Thus, it is not correct to say that acylation is "required for TCR recruitment" (line 4, 12), or that Orai C143A "fails to reach the immune synapse" (line 58-9). These kinds of absolute statements should be changed throughout the text to better reflect what the data show.

---

## [Author Response]

[Editors’ note: the authors resubmitted a revised version of the paper for consideration. What follows is the authors’ response to the first round of review.]

1. Measure the expression of C143A Orai1 in the PM, to address the possibility that C143A reduces Orai1 activity through altered trafficking.

We thank the reviewer for this comment. We have used 3 strategies to measure the expression of C143A Orai1 in the PM.

1) We have used surface biotinylation to measure membrane exposure of WT or C143A ORAI1-YFP stably expressed in ORAI1-deficient Jurkat cells. As shown in the new Figure 5B, the biotinylated fractions contained similar amounts of ORAI1-YFP and STIM1 relative to the surface marker PMCA.

2) We have quantified the proportion of TIRF vs. total YFP fluorescence in our HEK-S1/O1 cells stably expressing WT or C143A YFP-ORAI1. As shown in the new Figure S3B, the YFP fraction detected in the TIRF plane was similar for these two constructs over a large range of expression levels.

3) We have used a YFP-ORAI1 construct bearing a HA tag in the ecto domain (kindly provided by Khaled Machaca). A C143A mutant was generated, the constructs expressed in HEK-293 cells, and HA immunoreactivity and YFP fluorescence measured by confocal imaging. As shown in Figure S2, the ratio of HA over YFP fluorescence was comparable in non-permeabilized cells expressing WT and C143A HA-tagged ORAI1, indicating comparable surface expression.

These 3 independent assays establish that C143A and WT ORAI1 have comparable steady-state PM levels and thus that the cysteine substitution does not reduce channel activity via altered trafficking.

2. Compare the SOCE responses of WT and C143A Orai1 after knocking down the various PATs, to determine whether C143A reduces Orai1 activity through acylation-independent mechanisms.

We have compared the SOCE responses of WT and C143A HEK-S1/O1 cells depleted of the three PAT isoforms. As the residual SOCE of C143A cells is very low, we increased the amount of readded Ca^2+^ from 0.5 to 1 mM to increase the dynamic range of the response. As shown in the new Figure S4C-D, siRNA-mediated depletion of either PAT-3, 7, or 20 does not inhibit SOCE in C143A cells. Instead, the residual SOCE responses were slightly increased by silencing the PAT isoforms. This indicates that C143A does not reduce ORAI1 activity through acylation-independent mechanisms.

3. Obtain clearer data to characterize Orai1 WT and C143A targeting to lipid domains.

We are very thankful for this suggestion which we believe has increased the quality of the study. To obtain clearer data, we used a PIP2 sensor instead of DiD to mark non-raft domains and a new batch of cholera toxin B coupled to Alexa 647 to mark raft domains. With these new markers, the different lipid fractions were clearly separated, allowing to quantify the distribution of the YFP-tagged channels in raft vs. non-raft domains from opposite poles of the same vesicle, according to {10.1038/nprot.2012.059 }. As shown in the new Figure 3D, these experiments revealed that ORAI1 accumulates preferentially in PIP2-rich domain, with a minor fraction localized in raft domains. This fraction was further reduced in cells expressing the C143A mutant, but the difference was not significant.

We therefore postulated that the basal level of palmitoylated ORAI might be low, and overexpressed PAT20 to maximize the channel palmitoylation. As shown in the new Figure 4D, PAT20 overexpression increased the fraction of WT ORAI1-YFP located in lipid rafts, without altering the distribution of C143A ORAI-YFP. We believe that these results convincingly show that PAT20-mediated palmitoylation of Cys143 modifies the lipid preference of the ORAI1 channel to promote its accumulation in lipid rafts and thank the reviewers for prompting us to improve the lipid assays.

4. Resolve the inconsistent effects of up- and downregulating PATs to clarify which PATs are responsible for Orai1 S-acylation.

We now show that the mRNA levels of PAT20 are decreased by the silencing of the two other isoforms (Figure S4A). This off-target effect of the siRNAs likely accounts for the inconsistent effects of up- and downregulating different PATs. We also show that these 3 siRNAs do not further reduce the activity of ORAI1-C143A (see response to editorial comment 2), while overexpression of any of the 3 isoform increases the S-acylation of WT, but not of C143A ORAI1. The three enzymes therefore S-acylate ORAI1 on a single residue, Cys143, but PAT20 is the only isoform associated with a loss or gain in channel function. The most likely explanation therefore is that ORAI1 undergoes successive cycles of Sacylation, involving PATs 3, 7 and 20, and of de-acylation. We confirmed that ORAI1 is de-acylated by pulse-chase experiments, using an inhibitor of APT enzymes, in cells expressing the different PATs (Author response image 1). PAT20 overexpression was the most efficient in preventing the depalmitoylation, suggesting that the lipid modification mediated by this enzyme is more stable. PAT20 might be the rate-limiting enzyme in successive cycles of S-acylation or could act at a preferential location such as the plasma membrane to transiently S-acylate ORAI1. This is consistent with the recent report that ORAI1 is rapidly and transiently S-acylated in Jurkat T cells exposed to CD3 beads. {https://pubmed.ncbi.nlm.nih.gov/34156466/}. Our data therefore establish that PAT20-mediated Sacylation at Cys143 enhances ORAI1 channel function.

**Author response image 1. sa2fig1:** CRISPR ORAI1 KO + ORAI1 WT Jurkat cells were treated with ML349 (10µM 24h) or untreated (left) or transfected with PAT3, PAT7 or PAT20 (Right). Cells were pulsed for 2 H with palmitate and chased for the indicated times. Autoradiograms show IP of ORAI1-YFP with GFP beads.

5. Obtain more consistent data relating Ca signals, NFAT translocation and IL-2 secretion in Jurkat cells.

We thank the reviewers for pointing out the discrepancies in the environmental conditions of our previous assays. As requested, we have repeated key calcium experiments at 37°C to mimic the conditions of the NFAT translocation and IL-2 assays. As shown in the new Figure 5A, the SOCE response recorded at 37°C was abrogated in ORAI1-CRISPR Jurkat cells reconstituted with ORAI1C143A.

We have also used a NFAT promoter fused to luciferase normalized to *Renilla* to directly record NFATdependent transcription. Tg failed to activate NFAT transcription in ORAI1-CRISPR Jurkat cells reconstituted with the C143A mutant (new Figure 5D).

Importantly, reduced Ca^2+^ signals were observed in Jurkat T cells activated with CD3 beads at 37 °C (new Figure 5C).

These new data show that Ca^2+^ entry, NFAT activation, and Il-2 production elicited by Tg and by TCR engagement are equally compromised in Jurkat cells reconstituted with the C143A mutant. At 37°C, the alterations in Ca^2+^ signals match the defects in downstream Ca^2+^-dependent pathways. We thank the reviewers for prompting us to repeat these experiments in identical physiological conditions.

6. Perform the immune synapse experiments under conditions that will allow conclusions about physiological effects of acylation on immune cell activation. While conducting studies with primary T cells would be preferable, the reviewers agreed that Jurkat cells could be used as a first model here, but only at a physiological temperature that fully supports synapse formation and with a more physiological substrate (e.g., supported lipid bilayers) that would allow movement of TCR complexes in the plasma membrane. These experiments would likely be the most time-consuming, but are needed to support the impact of S-acylation on immune function, a main area of significance for the paper.

We thank the reviewers for pointing out the limitations of our immune synapse assay. We agree that the use of CD3-coated coverslips at room temperature is not optimal to study immune synapse formation. As suggested, we have repeated these experiments using a more physiological substrate and temperature. We used RAJI cells pulsed with *Staphylococcus aureus* enteretoxin E (SEE), an established model to study immune synapse formation in Jurkat cells (doi.org/10.1074/jbc.M109.097311, doi: 10.3791/60312). We first validated immune activation downstream of IS formation using the NFAT luciferase assay. As seen in the new Figure 6B, 24h of co-culture (1:1 ratio) of SEE-pulsed RAJI induced a strong NFAT luciferase activity in Jurkat reconstituted with ORAI1-WT, while no response was observed in ORAI1-C143A cells. ORAI1 acylation is therefore also required for the proper activation of Jurkat T cells in this physiological immune synapse assay.

We further established that acylation also promotes ORAI1-YFP accumulation in these IS by live cell imaging. As shown in the new Figure 6C-E, WT ORAI1-YFP accumulated in the IS in 90 % of cells while C143A ORAI1-YFP was enriched in the opposite pole, known as distal cap, in up to 60% of cells. Time-resolved recordings revealed that WT ORAI1-YFP accumulated within 10 min in the IS while C143A ORAI1-YFP remained evenly distributed between the IS and distal caps throughout the recordings.

We also quantified the distribution of actin and of TCR in IS by confocal imaging. Jurkat and SEE-pulsed RAJI cells (stained with cell mask) were seeded on fibronectin coverslips a 1:1 ratio for 1h, fixed, and stained for anti-TCR-PE and Phalloidin 405. As shown in the new Figure 6F-G, IS forming in Jurkat T cells reconstituted with the C143A mutant contain less ORAI1-YFP, les actin, and less TCR, consistent with our earlier findings in CD3-coated activating coverslips. These data indicate that S-acylation of ORAI1 sustains IS formation and TCR activation during physiological IS formation.

We believe that our new data using RAJI cells establish that ORAI1 S-acylation is required for efficient immune function and thank the reviewers for prompting us to repeat the immune synapse experiments in conditions that allow movement of TCR complexes in the plasma membrane.

Reviewer #1:[…]Essential revisions:1. The evidence for acylation at C143 is clear, but I am not convinced a single enzyme (PAT20) is responsible for it.

As discussed in our response to the editor (point 4), our data show that at least three enzymes, PAT3, 7, and 20 can S-acylate ORAI1 on a single residue, Cys143. We have also performed palmitate incorporation studies in Jurkat CRISPR KO ORAI + ORAI1 WT and transiently expressed all PATS (Author response image 2). In this settings, only PAT3, 7 and 20 increase ORAI1 S-acylation.

**Author response image 2. sa2fig2:** CRISPR ORAI1 KO + ORAI1-WT Jurkat cells transiently expressing each of 23 PAT isoforms were labeled with tritiated palmitate. Autoradiograms show IP of ORAI1-YFP with GFP beads. Only PAT3, 7 and 20 promoted ORAI1 S-acylation.

a) Overexpression of a number of PAT proteins can increase 3H-palmitate incorporation in two bands which do not migrate at the same MW as Orai (Figure 4A). Either this is an error in labeling the gel, or another protein that co-ip's with Orai1 is the target.

Indeed, the MW labels were not properly aligned on the bottom gel of Figure 4A. We apologize for this. The 3H-palmitate is detected in two bands that migrate at the same MW as the ORAI1-YFP fusion protein. Identical results were obtained with PAT overexpression in Jurkat T cells reconstituted with ORAI1-YFP (new Figure S5E and S5F).

b) There is no evidence that the doublet of bands corresponds to Orai1alpha and Orai1beta, and the authors' claim that PAT20 only acylates the high MW band while PAT3 and PAT7 acylate the lower MW band is not strictly true. All three acylate both forms, with a slight preference for one over the other. To make this conclusion, the authors would need to express the α or β forms of Orai1 separately and show differential effects on acylation.

We agree that the two bands exhibit minor differences in S-acylation and that our proposal was speculative. We have since repeated the S-acylation experiments in Jurkat T cells overexpressing the different PATs. These data (new Figure S5E and S5F) show that all 3 PATs increase 3Hpalmitate incorporation in two bands that co-migrate with ORAI1-YFP. PAT20 therefore does not preferentially acylates one of the two bands and we have modified the text accordingly.

c) It seems inconsistent that knockdown of all 3 PATs (3, 7, and 20) reduce SOCE equally (Figure S4B), yet only overexpression of PAT20 enhances SOCE (Figure 4), particularly since PAT3 and PAT7 were not knocked down to nearly the same extent as PAT20 (Figure S4B) and PAT20 was hardly overexpressed (Figure 4B).

We now show that the mRNA levels of PAT20 are decreased by the silencing of the two other isoforms (Figure S4A). This off-target effect of the siRNAs likely accounts for the inconsistent effects of up- and downregulating different PATs. We also show that these 3 siRNAs do not further reduce the activity of ORAI1-C143A (see response to editorial comment 2), while overexpression of any of the 3 isoform increases the S-acylation of WT, but not of C143A ORAI1. The three enzymes therefore S-acylate ORAI1 on a single residue, Cys143, but PAT20 is the only isoform associated with a loss or gain in channel function. The most likely explanation therefore is that ORAI1 undergoes successive cycles of S-acylation, involving PATs 3, 7 and 20, and of de-acylation. We confirmed that ORAI1 is de-acylated by pulse-chase experiments, using an inhibitor of APT enzymes, in cells expressing the different PATs (Author response image 1). PAT20 overexpression was the most efficient in preventing the depalmitoylation, suggesting that the lipid modification mediated by this enzyme is more stable. PAT20 might be the rate-limiting enzyme in successive cycles of S-acylation or could act at a preferential location such as the plasma membrane to transiently S-acylate ORAI1. This is consistent with the recent report that ORAI1 is rapidly and transiently S-acylated in Jurkat T cells exposed to CD3 beads. {https://pubmed.ncbi.nlm.nih.gov/34156466/}. Our data therefore establish that PAT20mediated S-acylation at Cys143 enhances ORAI1 channel function.

d) The partial overlap of Orai1 and PAT20 fluorescence in the cell does not add significant support to the role of PAT20 in acylating Orai1 (Figure 4D).

We agree and have now removed this from the revised version of the manuscript.

2. The reduction of Orai1 activity by the C143A mutation is clearly demonstrated, but the effects on activation kinetics and inactivation are not. The activation time course could be simply measured by time to half peak or 90% peak current; I am not sure what shifting the traces in time to align them by their "inflection points" signifies physiologically. With regard to inactivation, are the authors referring to Ca^2+^-dependent inactivation, and if so, how would this occur in the presence of 10 mM BAPTA in the pipette? A mechanistic description of this is not necessary, but more examples of the inactivating currents should be shown in a supplementary figure, with an average time course to characterize the overall effect.

We agree that the kinetics differences between WT and C143A ORAI1 were not properly reported. We have modified the graph as suggested to show average time courses and include original recordings as supplementary material. The activation time course is now reported as time to 90% peak current without alignment to inflection points (Figure 2F and S2B), which was significantly increased in cells expressing the C143A mutant. We now show more examples of currents inactivating in WT cells perfused with 10 mM BAPTA (Figure S2A) but have removed the statistial evaluation of this inactivation whose mechanistic basis is unclear as the interpretation of kinetics differences is complicated by the small amplitude of C143A currents.

3. Does C143A alter Orai1 activity only by preventing acylation, or could it have acylation-independent effects? If it is only due to prevention of acylation, then eliminating PAT activity should have the same effect in cells expressing WT Orai1 (Figure S4B) and Orai1 C143A. This comparison should be made.

We have performed the requested experiments and compared the effect of PAT silencing in cells expressing WT Orai1 and Orai1 C143A. We believe that the reviewer’s comment is incorrectly worded however, as eliminating PAT activity should not impact ORAI1-C143A channel activity if this residue only act by preventing acylation. As discussed in our response to point 1c above, downregulating any of the 3 PAT enzymes involved in ORAI1 S-acylation did not further reduce the activity of ORAI1-C143A. This indicates that C143A alter Orai1 activity by preventing S-acylation.

4. To pursue a mechanism to explain how C143A reduces Orai1 activity, the authors describe effects on diffusion rate, Orai1 puncta size, and loss of partitioning into cholesterol-rich domains. However, these data do not provide clear support for a unifying mechanism.a) Diffusion rate. The FRAP experiments are nicely done, but the slowing of diffusion is slight (~15% in Figure 3C). It is not clear how this would lead to a large change in coupling to STIM1 as hypothesized in the Discussion (lines 236-238).

The slowing of diffusion is slight, but the difference is significant, and persisted after restricting the analysis to cells with similar levels of YFP-ORAI1 fluorescence. We now show the primary data for both WT and mutant ORAI1 (new Figure 3C and S3D). The reduced mobility is consistent with the preferential localization of C143A-ORAI1 in non-raft lipid domains. This supports our mechanistic model that S-acylation target ORAI1 to PM domains devoted to local Ca^2+^ signaling like the immune synapse. Whether the slightly reduced mobility in lipids translates into a reduced trapping by STIM1 at ER-PM contact sites would require single-molecule analysis, which is beyond the scope of this study. We have rephrased our discussion to acknowledge these points.

b) Orai1 puncta size. The time course of Orai1 cluster size (Figure S3) is puzzling and inconsistent with published data. Even with WT Orai1, size continues to increase slowly well after STIM1 has reached a plateau. In previous reports on HEK cells, Orai1 accumulates only slightly (~10 s) after STIM1 (e.g., Figure S3 of Perni et al., PNAS 112:E5533, 2015). It does not make sense that Orai1 cluster size would continue to increase for a long time, when STIM1 size is constant, considering that Orai1 clusters by binding to STIM1. Perhaps this is a problem with the method used to analyze Orai1 puncta fluorescence? A more direct and interpretable approach would be to measure by TIRF the level of Orai1 (and STIM1) fluorescence in the cell footprint over time. Does C143A reach a lower plateau fluorescence, hence a reduced number of channels able to conduct Ca^2+^? Perhaps this could be explained by diminished coupling to STIM1.

We believe that the differences in kinetics stem from differences in the analysis of the TIRFM data. In the Perni et al. study, STIM1 puncta were used as a mask and fluorescence values normalized to the range of final minus initial fluorescence to report the kinetics of mCh-ORAI1 accumulation at the sites of puncta formation. We segmented every cell and fluorescence channel independently at each time point to determine the variations in the size of individual clusters. The continuous increase in the size of ORAI1-YFP clusters in our time-course is due to the use of a fixed threshold set at a high value, based on the intensity of the last image. We realize that this approach can be misleading and have re-analyzed the data using an adaptive threshold. Our re-analysis revealed also that we were comparing cell with different basal intensities, since we threshold based on fluorescence we decided to restrict the cells analyzed to comparable fluorescence levels (Author response image 3). The new analysis (Figure 3B and S3C) shows that WT and C143A ORAI1 accumulate with similar kinetics in clusters and confirms that C143A ORAI1-YFP forms fewer and smaller clusters. As requested, we also report the changes in fluorescence in the cell footprint over time. The mCh-STIM fluorescence increased 5-fold while the ORAI1-YFP increased marginally, consistent with the re-localization of ORAI1 channels within the PM.

**Author response image 3. sa2fig3:** ORAI fluorescence in HEKs1/O1 WT or C143A in our previous submission and in the new one, with fluorescent matched basal levels.

c) Lipid partitioning. I have concerns about the partitioning of Orai1 WT and C143A in raft and non-raft domains based on the quality of the images in Figure 3D. Why is the distribution of DiD and CxB so different in the upper and lower rows? C143A seems to have a non-uniform distribution that does not correspond to either the CxB or DiD distributions. The method of analysis is not described, and the different categories in the bar plot in Figure 3D are not explained. The authors should supply control data to show that the method yields meaningful results.

Please refer to our answer to the editorial summary, point 3.

5. To explain the differences in Ca^2+^ signals in Jurkat cells expressing WT or C143A Orai1, it is imperative to know whether they arise from altered expression at the cell surface vs. altered properties of the channels themselves. Acylation has been shown to affect many aspects of ion channel biosynthesis, maturation, trafficking and localization as well as function. FACS analysis of cells stained with anti-Orai1 antibody can be used to measure surface expression (with appropriate controls). and a comparison of surface labeling with total GFP fluorescence (e.g. Figure S5D) can then be used to determine whether acylation affects trafficking to or from the plasma membrane.

Please refer to our answer to the editorial summary, point 3.

Were all copies of the endogenous Orai1 gene eliminated in the CRISPR Jurkat cells?

The disruption of the endogenous Orai1 gene was verified by genomic sequencing. We now include a WB showing the absence of ORAI1 immunoreactive band in CRISPR Jurkat cells (new Figure S6B).

6. The authors conclude that acylation is required for efficient NFAT translocation to the nucleus, but the supporting data are inconsistent. With TG, KO of Orai1 causes only <50% decrease in NFAT translocation (Figure 5D), whereas a nearly complete suppression would be expected based on the SOCE response, and C143A almost completely rescues SOCE (Figure 5A), yet does not rescue NFAT translocation at all relative to the KO (Figure 5D). In response to anti-CD3, C143A does not promote detectable NFAT translocation, yet it stimulates IL-2 production by ~50% relative to WT Orai1 (Figure 5F). Were these experiments done at the same temperature? The conditions are not adequately described in the Methods. The lack of internal consistency of these results lessens confidence in the conclusion that Orai1 acylation is required for T cell transcription by TCR engagement.

Please refer to our answer to the editorial summary, point 5.

7. The authors conclude that C143A Orai1 interferes with actin accumulation, Orai1 localization, and TCR cluster formation at the immune synapse. There are several problems with the data and experimental design that undermine this conclusion.a) The proper formation of an artificial synapse on anti-CD3 coated coverslips is highly temperature dependent and does not occur efficiently at room temperature, the condition under which these experiments were done. For this reason, IS studies are always conducted at 37 deg C (see papers by Bunnell, Hammer, Dustin groups). Because low temperature may exacerbate effects of reduced Ca^2+^ entry on the synapse, it is difficult to use these results to infer how much acylation might affect synapse formation under physiological conditions.b) The quality of the actin images is low in comparison to published work, making it difficult to judge the degree to which its structure has been changed by C143A Orai1. Given the quality of the actin images, it is hard to tell how actin rings and center, middle, and peripheral regions (cSMAC, pSMCA, dSMAC?) were detected and quantitated (Figure S6). Generally, actin filament organization in living cells is described using labeled probes like LifeAct or F-tractin (e.g., Yi et al., Mol. Biol. Cell 23:834, 2012). One of these probes should be tried, as SiR-actin could interfere with normal actin dynamics due to its structural similarity to jasplakinolide.c) How do TCRs rearrange to accumulate in the center of the contact zone when they are bound to surface-attached, immobile anti-CD3, and in the movies why do the PE-labeled anti-TCR antibodies appear to move (the time scale should be stated)? Are we looking at internalized TCR?

As suggested, we have repeated the immune synapse experiments in conditions that allow movement of TCR complexes in the plasma membrane, at physiological temperature, using phalloidin to stain actin. Please refer to our answer to the editorial summary, point 6.

Reviewer #2:In previous studies, Orai1 was identified as a highly S-acylated protein in human T cells, as well as other murine neural stem cells. The main finding of this study is that S-acylation of Orai1 on Cys143 by PAT20 (or ZDHHC20) regulates CRAC function, lipid mobility and ultimately cytokine production by T cells upon TCR stimulation. Using acyl-PEG exchange, 3H-palmitate incorporation and mutagenesis of cystein residues, the authors show that Orai1 is S-acylated on Cys143, a highly conserved Cys residue, in HeLa cells. Authors show that PAT20, expressed in T cells, can S-acylate Orai1 and modulate SOCE. The authors then investigated the impact of Cys143 S-acylation on CRAC channel function by calcium imaging and patch clamping in HEK-293 and Jurkat cells and show a strong reduction in SOCE and ICRAC in cells expressing a C143A mutant (and thus S-acylation defective) form of Orai1. Furthermore, Orai1-C143A shows reduced mobility in membrane lipids, altering its accumulation in lipid rafts and efficient Orai1 clustering upon CRAC activation. Finally, authors demonstrate that S-acylation of Orai1 is required for robust calcium responses upon TCR stimulation in Jurkat cells leading to NFATc1 translocation and IL-2 production.In Figure 3C, the effects of the Orai1-C143A mutant on lateral diffusion are very small. While statistically significant, it is unclear whether this difference is really meaningful. It would be useful if the authors could show the primary data for mutant Orai1 in addition to Orai1 WT.

We have reanalyzed the FRAP data to confirm that the difference in lateral diffusion persists in cells with similar levels of YFP-ORAI1 fluorescence. We now show the primary data for both WT and mutant ORAI1 (new Figure 3C and S3D). The reduced lateral mobility is consistent with the preferential localization of C143A-ORAI1 in non-raft lipid domains.

The conclusion from Figure 4 that PAT20 S-acylates Orai1 is plausible, but the argument could be strengthened by performing palmitate incorporation in cells expressing Orai1 WT or Orai1-C143A together with PAT isoforms. This is shown only indirectly through measurements of SOCE in panel C.

We have measured palmitate incorporation in Jurkat KO cells reconstituted with WT or C143A ORAI1-YFP and transiently transfected with the 3 PAT isoforms. These data (New Figure S5F) clearly show that PAT3, 7 and 20 S-acylate ORAI1 on Cys143.

Regarding the effects of PAT20 on SOCE shown in Figure S5F,G, I have a similar comment as in Figure 4: repeating this experiment with Orai1-WT and Orai1-C143A overexpression would strengthen the authors conclusions that PAT20 directly regulates Orai1 to modulate SOCE and IL-2 production instead of indirectly through acylation of another protein.

We did not measure SOCE in Jurkat C143A cells overexpressing PAT20 as we believe that this experiment will not establish whether PAT20 acts directly or indirectly to regulate ORAI1. Our data establish that at least 3 PATs modulate SOCE via the S-acylation of ORAI1 Cys143 residue. As discussed in our response to the editorial summary, point #4, we believe that the Cys143 residue is reversibly S-acylated by these enzymes during successive cycles of S-acylation and de-acylation. We confirmed that ORAI1 is de-acylated by pulse-chase experiments, using an inhibitor of APT enzymes (see answer to Reviewer 1 comment 1c). The gain of function conferred by PAT20 overexpression suggests that PAT20 is the only rate-limiting enzyme in cycles of successive S-acylation, but since the C143A mutant is not acylated it cannot be used to establish whether PAT20 acts directly or indirectly through acylation of another protein. To establish this point would require biochemical assays that are beyond the scope of this manuscript.

Figures 6A,B: From the representative images and quantifications shown I am not at all convinced that Orai1-WT localizes to the bead contact are (which the authors call immune synapse). If that were the case, why does the percentage of ORAI1-GFP at cups not increase over time (from 0 to 8 min)? The gray trace in Figure 6B is flat! The apparent difference between WT and C143A is only seen at one time point (8 min after stimulation) and results from reduced C143A at the cups. The quantification in Figure 6B is based on 5 and 3 cells expressing Orai1-WT and Orai1-mutant, respectively, which is not sufficient to draw these conclusions. Since the role of Orai1 acylation in IS formation and signaling features prominently in the title and abstract of the paper, these conclusions need to be solidified. (On a side note, looking at the data in the referenced papers (e.g. Ref. 21, Lioudyno et al), they are not that convincing either to support a recruitment of Orai1 to immune synapses in my opinion).

Please refer to point #6 of the editorial summary.

The most surprising finding of this study is that Orai1-C143A lacking acylation and promoting reduced SOCE impairs TCR clustering. This argument is entirely based on the anti-CD3 coating condition, and may as well be an artifact. Assuming Figure 6D-F show the quantification of data in 6C, it would be misleading to label the bar graphs as "Orai1-GFP at IS". Since bound anti-CD3 antibodies do not diffuse (as for instance GPI anchored MHC molecules in lipid bilayers), a proper IS cannot form in T cells. I would suggest to confirm this finding using another system to measure immune synapses, either the CD3 bead system used in 6A-B or even better lipid bilayers. The authors discuss a number of explanations in the Discussion of their paper, but these explanations or the hyposthesis itself that Orai1/SOCE regulates TCR clustering are not represented in the graphical abstract in 6G.

As suggested, we have studied immune synapse formation in conditions that allow movement of TCR complexes in the plasma membrane, using SEE-pulsed RAJI cells at physiological temperature (see answer to editorial summary, point 6). The new data confirm our earlier findings that defective ORAI1 S-acylation impairs IS formation, T cell activation and TCR signalling. We cannot study TCR clustering in this setting and we have removed this statement from the text, and changed it for signaling. This is now indicated in the graphical abstract.

To make conclusions about the role of Orai1 acylation in T cell function, I would suggest to not only rely on a lymphoma cell line (Jurkat) but to confirm at least some of the key findings in real T cells. Alternatively, it would be appropriate to change the title and abstract of the paper.

In keeping with the recommendations of the senior editor, we use Jurkat cells as first model in this study and have changed the title and abstract accordingly.

Figure S4B: how do the authors explain that PAT20 (but not PAT3 or PAT7) overexpression increases SOCE, whereas siRNAs targeting PAT20, PAT7 or PAT3 all decrease SOCE to a similar extent? Are siRNAs specific to these isoform?

Please refer to our answer to point #4 of the editorial summary. We now show that the mRNA levels of PAT20 are decreased by the silencing of the two other isoforms (Figure S4A). This offtarget effect of the siRNAs likely accounts for the inconsistent effects of up- and downregulating different PATs and we thank the reviewer for prompting us to properly carry out this important control.

Figure 5 and S5: I am not quite convinced of the NFAT localization data. Are the graphs in Figure 5D the quantification of data in Figure S5E? How long were cells in 5D stimulated with thapsigargin? What does "NT" mean in S5E? In the representative data (?) in S5E, I do not see a difference in nuclear localization of NFATC1 in WT and mutant Orai1 transfected cells. How do the authors explain the discrepancy between Figure 5D and E with regard to (a) the NFATC1 nuclear /cytoplasmic ratio (approx. 6 in Orai1-WT cells after TG in D, but approx. 2 in Orai1-WT cells after CD3) and (b) the only small defect in D but complete defect in E?

Please refer to our answer to point #5 of the editorial summary. We now complemented these experiments with an NFAT promoter fused to luciferase and a control plasmid encoding the *Renilla* luciferase to directly record NFAT-dependent transcription (new Figure 5D). Regarding the endogenous NFAT nuclear translocation, NT means not treated, this is now explained in the figure legend. TG was used for 4 hours while CD3 for 30 and 120 minutes. We believe these 2 conditions have different dynamic ranges because Tg responses are much robust and sustained than CD3. To support this, we have now performed a line scan of NFAT and nuclear signal after 4 hours to observe the differences in endogenous NFAT re-localization. The small defect in D but big difference in E is consistent with the recent report that ORAI1 is rapidly and transiently S-acylated in Jurkat T cells exposed to CD3. {https://pubmed.ncbi.nlm.nih.gov/34156466/}.

Reviewer #3:[…]1) The assessment of Orai1 expression in the plasma membrane is an important variable and rigorous tests need to be done to check whether this key parameter is affected. In Figure S1C, the authors show widefield images of mCh-STIM1 and wildtype Orai1-GFP. The images seem to be low quality and it is difficult to assess whether mCh-Orai1 is significantly affected in the C143A mutant. What exactly was quantified in the summary bar graph next to the image? Plasma membrane expression? Total cell expression? How were the ROIs drawn? As a moderate drop in Orai1 expression could easily explain the reduction in SOCE and CRAC currents, this key parameter (Orai1 plasma membrane expression) needs to be determined with much greater precision to rule it out. Importantly, the quantification should be done using confocal microscopy and backed by additional confirmatory tests such as biotinylation of membrane proteins to determine whether plasma membrane Orai1 fraction is affected.

As discussed in our answer to point #1 of the editorial summary, we have used 3 strategies to rigorously assess the expression of Orai1 in the PM: surface biotinylation, TIRF vs. total YFP fluorescence, and surface immunoreactivity of epitope tagged ORAI1 constructs. These 3 independent assays establish that WT and mutant ORAI1 have comparable surface expression and thus that the cysteine substitution does not reduce channel activity via altered trafficking.

2) STIM1-Orai1 binding. It is unclear whether C143A affects STIM1 binding. One could envision that impaired Orai1 accumulation at the ER-PM junctions and ensuing changes in SOCE could result from impaired binding to STIM1. This should be assessed using STIM1-Orai1 FRET and/or other methods.

We have assessed STIM/ORAI binding by co-immunoprecipitation in HEK S1/O1 cells treated for 15 min with Tg to promote binding. As shown in Author response image 4, we observed comparable levels of binding between mCh-STIM1 and WT or mutant ORAI1 constructs. The cysteine substitution thus does not cause major alterations in STIM1-ORAI1 binding at the cellular level, as reported recently in (https://doi.org/10.1242/jcs.258579). We did not include these data in the revised manuscript as we believe that local rather than global interactions would be more relevant to study but feel that such experiments are beyond the scope of our current study.

**Author response image 4. sa2fig4:** 

3) It would be interesting to determine whether C143A affects fast Ca^2+^ -dependent inactivation. A lack of affect would provide further support for the idea that the main change is the number of active Orai1 channels activated rather than a change in Orai1 channel properties.

As suggested, we have measured FCDI in HEK-S1/O1 cells. To our surprise, we observed a more rapid current deactivation followed by a reactivation in cells expressing C143A ORAI1 and perfused with 10 mM EGTA during hyperpolarizing pulses ranging from -120 to -60 mV (new Figure S2C). The fractional current at the end of the 200 ms pulse was increased by 25%, indicating that the cysteine mutant exhibits reduced overall FCDI. We do not know the mechanistic basis of this alteration, which might be related to differences in the local lipid environment or in the local STIM1/ORAI1 ratio.

4) Figure 3C: The diffusion rate differences between WT Orai1 and C143A Orai1 seem rather small. I am concerned that the difference is mainly driven by the one outlier data point in the WT group that shows very high D (~30 µm2/s). Please remove this outlier and run the test again to determine whether the data is significant. It is difficult to see how this relatively modest difference accounts for the very large difference in Figure S3A between WT and C143A Orai1 clusters at the late time points.

We have reanalyzed the FRAP data to confirm that the difference in lateral diffusion persists in cells with similar levels of YFP-ORAI1 fluorescence (the difference also persisted after removing only the outlier data point). We now show the primary data for both WT and mutant ORAI1 (new Figure 3C and S3D). To address point #4b of Reviewer 1, we have also reanalyzed the TIRFM data and now report the kinetics of ORAI1-YFP cluster formation (Figure S3C). Whether the reduced lateral mobility of the mutated channel in lipids translates into a reduced trapping by STIM1 at ER-PM contact sites would require single-molecule analysis, which is beyond the scope of this study. We have rephrased our discussion to acknowledge these points.

5) Related to the above points, I am also unclear on why HEK cells expressing both STIM1 and Orai1 were used for these experiments. If the goal is to assess the diffusion of Orai1 uncontaminated with binding to other proteins (such as STIM1), isn't the experiment best done in cells expressing only Orai1?

Co-expression of STIM1 and ORAI1 is required to record CRAC currents and to track the formation of ORAI1-YFP clusters by TIRFM. We therefore generated HEK-293 cells stably expressing both proteins to characterize the functional defects associated with S-acylation deficient ORAI1-C143A. We then used the same two cell lines for the FRAP experiments to relate the functional defects to an altered mobility in membrane lipids. Of note, the FRAP experiments were performed in resting, store replete cells to minimize the recruitment of STIM1 to ER-PM contact sites. We are therefore essentially exploring the mobility of ORAI1 unbound to STIM1 in these experiments.

6) Figure 3D. The assessment of Orai1 distribution in rafts vs non-raft appears to have been carried out purely from widefield images. It is really difficult to see how this mode of imaging provides clear results. I would strongly recommend redoing these tests with confocal microscopy to improve the image quality and hence the robustness of the data.

Please refer to our answer to the editorial summary, point 3.

7) Figure 4A should be quantified using densitometry analysis. In this blot, the levels of tritiated palmitate seem substantially lower (and barely present!) than what is shown in Figure 1. Why is that?

The autoradiograms have to be exposed for several weeks to detect the incorporation of tritiated palmitate. In Figure 4 the exposure time was reduced to better reveal the increased incorporation associated with the overexpression of PAT enzymes. We now provide a densitometry analysis of the autoradiograms in Figure 1, 4, and S5. Among an array of PAT exogenously expressed, only PAT3, 7 and 20 increased by ~2.5 fold the incorporation of tritiated palmitate into full-length ORAI1-YFP expressed in HeLa (n=1), RPE1 (N=2) and Jurkat T cells (N=2).

[Editors’ note: further revisions were suggested prior to acceptance, as described below.]

Essential revisions:1. Many conclusions in the abstract are either overstated or not shown in the paper. These need to be revised or deleted.a. "acylation of Orai1 is required for TCR assembly" (see also line 51). TCR assembly was not studied in this paper.

We have replaced “TCR assembly by “TCR recruitment”.

b. “The acylation-deficient channel accumulated in cholesterol-poor domains and failed to reach the IS, preventing proper IS formation.” The data show no significant effect of C143A on accumulation in cholesterol-poor domains (Figure 3D), and C143A reaches the IS, just not as well as WT (Figure 6D, E). Also, the synapses were not characterized in sufficient detail to make a statement about whether they are “proper.”

We have rephrased this sentence to: “The acylation-deficient channel remained in cholesterol-poor domains upon enforced ZDHHC20 expression and was recruited less efficiently to the IS along with actin and TCR”.

c. “S-acylation is a critical regulator of Orai1 channel assembly.” Orai1 channel assembly was not studied in this paper.

We have replaced “Orai1 channel assembly” by “Orai1 channel trafficking”.

d. "local Ca^2+^ fluxes are required for TCR recruitment to the synapse." Local Ca^2+^ fluxes were not measured, and TCR recruitment to the IS was not significantly altered with C143A (Figure 6G).

We have rephrased this sentence to “ORAI1 S-acylation is required for TCR recruitment to the synapse”. TCR recruitment to the IS was significantly altered with C143A as is now better appreciated in the new Figure 6G presenting these data as fold IS enrichment.

2. Because so many conclusions about the effects of acylation are based on C143A, it is essential to show that the behavior of C143A derives from the absence of acylation rather than a direct effect of the mutation on channel function. This could be done by comparing SOCE in C143A to WT+siPAT20; if effects are only due to inhibiting acylation, the two should be similar. The experiments of Figure S4B and C would allow such a comparison if the same [Ca^2+^] were applied in both.

We have performed the requested experiments. The new Figure 4 supplement 2 shows that cells expressing ORAI1-C143A and ORAI1-WT+siPAT20 have comparable SOCE amplitudes if the same [Ca^2+^] is applied in both. We previously used a higher Ca^2+^ concentration to facilitate the detection of low-amplitude SOCE of ORAI1-C143A cells.

Likewise, to rule out the possibility that the C143A mutant has downstream effects unrelated to Orai1 s-acylation, NFAT activation and immune cell function should also be tested using PAT20 knockdowns in addition to the C143A Orai1 analysis.

We have performed the requested experiments. The new Figure 5 supplement 7 shows that PAT20 silencing reduces IL-2 production on WT but not C143A Jurkat T cells and mimics the effect of the C143A mutation, linking the defect to the lack of ORAI1 S-acylation. These points are now discussed on lines 174-176.

3. The results show reduced accumulation of Orai1 in puncta, but analysis in terms of the size and number of Orai1 clusters makes it difficult to interpret. Cluster size and number is determined by the size and number of ER-plasma membrane junctions, which should not be affected by mutations in Orai1. In fact, STIM1 accumulation appears to be normal in C143A cells (Figure S3C). These results are probably due to a thresholding artifact; as intensity declines, the thresholded size and numbers of ROIs will also decline. The authors should instead measure the level of Orai1 accumulation in STIM1-containing puncta, which could be done by making a mask based on STIM1 puncta and applying it to Orai1 to get the average Orai1 fluorescence in puncta. This will indicate more directly the relative number of channels recruited to junctions.

We have reanalysed the TIRF data using the suggested procedure. The new Figure 3E shows that the average ORAI1-YFP fluorescence accumulating in STIM1-containing puncta is reduced in cells expressing the C143A mutant. We additionally show that the mutation reduces the Pearson’s co-localization coefficient between mCh-STIM1 and ORAI1-YFP (new Figure 3F). These data indicate that the number of channels recruited to STIM1-containing ER-PM junctions is reduced by the mutation that prevents ORAI1 S-acylation. The total ORAI1-YFP fluorescence was stable during cluster formation and the numbers of clusters reached a stable plateau (Figure 3-supplement 2), indicating that the difference in cluster size is not due to a thresholding artifact. We provide additional images to illustrate the reduced size of ORAI1-C143A clusters, which we found to be a prominent feature of the acylation-deficient ORA1 mutant (new Figure supplement 2).

4. It is not clear why only PAT20 overexpression enhances SOCE, when all three PATs (3/7/20) palmitoylate Orai1. The explanation given in line 228 about cycles of acylation and PAT20 being rate- limiting is not well developed. Why does only PAT20 enhance SOCE (Figure 4C) when all three proteins produce similar steady- state levels of palmitoylation (Figure S5E,F)?

It is indeed surprizing that only PAT20 enhances SOCE, when all three PATs (3/7/20) palmitoylate ORAI1 to the same extent. We have developed the concept that PAT20 is rate-limiting by proposing that the enzymes act at different locations to acylate the channel at different stages of its life cycle. This is consistent with the known subcellular location of these 3 PAT isoforms. PAT20 accumulates at the PM when expressed in HEK-293T cells, while PAT3 and PAT7 accumulate in perinuclear structures and colocalize with the Golgi marker GM130 (Ohno et al., 2006). We therefore propose that channel acylation in the Golgi is functionally neutral while S-acylation by PAT20 at the PM enhances channel function by impacting ORAI1 distribution in PM lipids. Our observation that PAT20 enhances the channel preference for cholesterol-rich domains of vesicles retrieved from the PM is consistent with this mechanism and with the enzyme acting at the PM. We now propose this mechanism to account for the different effects of the 3 isoforms (lines 242-246).

5. The C143A current develops more slowly than WT, but this is likely because the currents and [Ca^2+^]i increase produced by C143A are much smaller than WT and elicit much less slow Ca^2+^-dependent inactivation. The large WT currents reach a peak earlier because they begin to inactivate, and this does not seem to occur with C143A. This point should be discussed.

We now discuss the possibility that C143A currents develop more slowly due to a reduced Ca^2+^-dependent inactivation (lines 228-229).

6. The conclusion that C143A exhibits altered fast Ca^2+^ -dependent inactivation is intriguing but unconvincing. The initial phase of CDI in fact appears to be normal. What is different is the presence of secondary potentiation in C143A that is not seen in WT channels. Because of the way CDI is quantified (Iss/Ipeak), this metric would naturally be affected in the mutant. A previous study that looked at a cys-less Orai1 construct lacking all three endogenous cysteines concluded that CDI is qualitatively similar to WT Orai1 (PMID: 20018736), which would be consistent with the presence of inactivation that is also seen here. These results and conclusion should be restated to indicate that the mutant shows delayed time-dependent potentiation which clips off CDI that would otherwise have normally occurred (and not that CDI is impaired). [this is also consistent with the faster tau_fast in the mutant, Figure S2C]. More generally, the effects on CDI could arise from a direct effect of mutation or the lack of acylation on the CDI mechanism, or reduced STIM1 binding. Reduced affinity for STIM1 would be consistent with lower SOCE/ICRAC as well as reduced CDI. The authors should discuss these possibilities.

We agree with the reviewer’s interpretation of our electrophysiological data and have rephrased the corresponding sections to indicate that the C143A mutant exhibits a delayed time-dependent potentiation that masks an apparently normal FCDI. A similar pattern was observed with WT ORAI1 at STIM:Orai ratios below 1:2 (Scrimgeour et al., 2009) which is unlikely to occur in our conditions because we used a 2-fold higher amount of STIM1 plasmid in the transfection mixture to ensure a high STIM:Orai ratio. The ORAI1-C143A currents also had a positive reversal potential, indicative of a high STIM:Orai ratio (McNally et al., 2012). The molecular basis of the C143A potentiation is unclear as both the lack of acylation or the Orai1 mutation itself could impact STIM1 binding or affect the conformational transitions occurring within the channel during gating and inactivation. These points are now discussed in the revised manuscript (lines 101-103 and 228-231).

7. While C143A appears to diffuse a little more slowly than WT, it is hard to evaluate this effect because the D of WT Orai1 is much lower than reported by others (0.01 in this paper vs. ~0.08 μm2/s in ref 35 and Park et al., PMID 19249086). The authors should validate their method using a membrane protein with known diffusion coefficient; however, given the small magnitude of the effect, and uncertainty as to how it relates to the reduced activity of C143A, these data could be omitted.

We believe that the different values stem from differences in methodology and in the FRAP protocols. The diffusion coefficients reported by Wu et al. were obtained by single-molecule tracking of cells expressing low to moderate levels of tagged STIM1 and ORAI1. We performed FRAP experiments in cells stably expressing high levels of mCh-STIM and ORAI1-YFP. The diffusion coefficients reported by Park et al. were obtained by FRAP using large rectangular irradiating regions to bleach molecules across an entire cell axis. We bleached small circular regions inside cells based on a study using the same FRAP protocol which reported values very similar to ours for the diffusion of SOAR1 dimer–Orai1 complexes (Figure 2C and F of Zhou et al., 2018). Both the levels of expressed proteins and the geometry of the bleached regions could account for the differences, as the small circular regions are likely to be repopulated faster by neighbouring fluorescent proteins. The 27% reduction in the diffusion coefficient of the C143A channel is congruent with its reduced recruitment into PM clusters which might account for its reduced activity. We thus would like to keep these FRAP data on the final manuscript as supplementary material (new Figure 3 supplement 3).

8. It appears that the preference of Orai1 for rafts at 10 degC is not significantly affected by the C143A mutation, and the images do not show a noticeable difference either (Figure 3D). Both seem to be mostly excluded from CxB-labeled domains. The conclusion on line 146 is written as if Orai1 accumulation in rafts is somehow responsible for enhancing SOCE, but this seems unlikely given that C143A associates with raft lipids similarly to WT. Moreover, PAT20 has to be overexpressed to see a change in preference (Figure 4D), and yet the mean raft preference for WT+PAT20 in Figure 4D is the same as WT in Figure 3D, suggesting the measurements are not very reliable. Overall, given the small size of the C143A effect on partitioning, and the absence of a connection between raft localization and channel function, the relevance of these findings is limited and they could be omitted.

Quantifying the distribution of fluorescence proteins in giant plasma membrane vesicles is difficult as sample preparation and imaging must be performed at 10°C. Moreover, the analysis involves finding vesicles containing both CxB and PIP2-mCherry to extract fluorescence intensities from two opposite membranes. Small differences in temperature, protein expression levels, or culture conditions impact the measurements, rendering these experiments prone to inter-experimental variability. The reduced raft partitioning of cells expressing WT+Ctrl in Figure 4D compared to WT alone in Figure 3D might stem from the co-transfection of an additional empty plasmid, or from day-to-day differences in temperature or culture conditions. As requested, we have removed Figure 3D as the difference in raft partitioning is not significant (we believe due to the inter-experimental variability) but would like to keep Figure 4D to illustrate that enforced PAT20 expression enhances the preference of WT but not C143A Orai1 for lipid rafts (by 38%), a phenotype consistently observed in ORAI1-WT every experimental day (Author response image 5). We believe that this is an important information as it directly demonstrates that S-acylation at residue C143A enhances the preference of the ORAI1 channel for ordered lipid domains. Although we cannot directly link this altered lipid preference to channel function, earlier reports showed that translocation of the STIM1-ORAI1 complex between PIP2-poor and PIP2-rich domains impacts ORAI1 channel gating (Maleth et al., 2014). Our findings establish the molecular mechanism for a change in lipid preference and indicate that S-acylation can provide an additional level of regulation by controlling the channel distribution in lipids. We have rephrased our interpretation of the GPMV data to remove a direct link between raft localization and channel function (line 148).

**Author response image 5. sa2fig5:** Lipid raft preference of WT (left) or C143A (right) ORAI1-YFP co-expressed with an empty plasmid or PAT20. Related to Figure 4D. Data was aggregated by experimental day and paired to demonstrate that despite the inter-experimental variability, the raft preference of ORAI1 WT consistently increased when PAT20 is coexpressed (n = 4 independent days).

9. Figure 5F and G shows anti-CD3 fails to induce NFAT translocation with C143A Orai1, yet a substantial fraction of cells become IL-2 positive (2/3 of WT level). These results seem inconsistent given what we know about the role of NFAT in driving IL-2 expression.

We were also surprised that anti-CD3 failed to induce NFAT translocation in our Jurkat cells reconstituted with C143A Orai1. Since NFAT signalling relies on low and sustained calcium elevations (Dolmetsch et al., 1998) and Il-2 production integrates T cell signalling over 24h, we measured NFAT translocation at later time points. Our new Figure 5F shows that anti-CD3 induced a significant nuclear NFAT translocation at 4h in cells reconstituted with C143A Orai1. This delayed NFAT activation probably accounts for the 6% of C143A cells becoming Il-2 positive 24h after anti-CD3 treatment. Alternatively, Il-2 production by C143A cells might reflect the activation of Fos-Jun proteins that cooperatively bind with NFAT within the IL-2 promoter to activate transcription (Chinenov and Kerppola, 2001; Feske et al., 2000).

10. The conclusion (line 194-5) that "These data indicate that ORAI1 S-acylation is required to recruit the ORAI1 channel and TCR to form a signalling-competent immune synapse in Jurkat T cells" is not supported by the data. The effects of C143A on actin recruitment (<10%) and TCR recruitment (<5%) are extremely small, which would be more apparent if the data were plotted from 0 instead of 50 (Figure 6G).

We disagree with this comment, as the effects of C143A on actin and TCR recruitment are in fact large. The effect size was not well reflected on the original Figure 6G reporting IS enrichment as the fraction of the IS fluorescence relative to the total fluorescence of the two opposite membranes (F_IS_/[F_opposite pole+_F_IS_]). In this presentation, an even distribution of the fluorescent proteins at the IS and opposite membrane has a value of 50% and a maximal IS recruitment a value of 100%, thus we used 50 instead of 0 as minimal value. We now present the data as fold enrichment (F_IS_/F_opposite pole_) to better show the accumulation of fluorescent proteins at the IS (new Figure 6G). With this presentation, recruitment of actin and TCR to the IS is reduced by 38% and 55%, respectively, in cells reconstituted with ORAI1-C143A. We hope that the reviewers would agree that these effects are large and biologically relevant.

How are the Orai1 localization results in 6D/E and 6G related?

These are two distinct sets of experiments. Data in Figure 6D/E are derived from the time-lapse experiments in 6C. Data in Figure 6G are from cells fixed after 1h of co-culture with RAJI and stained with TCR-PE and phalloidin.

While the effects of C143A on SOCE, NFAT activation and IL-2 production are clearly significant, effects on the IS (TCR and actin accumulation) are not. Accordingly, the immune synapse data in Figure 6F-H and related parts of Figure S6 do not add significantly to the paper and could be deleted.

As discussed above the IS data on Figure 6F-G show a significant impairment of TCR and actin recruitment to the immune synapse of C143A cells. This defect is better appreciated with the data presented as fold IS enrichment. We believe that these data provide a novel and important insight on the role of ORAI S-acylation in sustaining IS formation and would like to keep Figure 6F-G in the paper.

11. Line 240: "Reduced ORAI1 trapping by STIM1 would reduce CRAC currents and possibly FCDI by increasing the amount of STIM1 molecules required to trap and gate deacylated channels". STIM1 binding to Orai1 C143A could be measured using FRET. This test should be considered as the CDI results suggest that C143A may affect gating, perhaps due to decreased STIM1 binding.

We unfortunately do not have access to STIM and ORAI fluorescent constructs for accurate FRET experiments (i.e STIM1-CFP stable cell line). As discussed in our reply to point 3 above, our new TIRF analysis now shows that the average ORAI1-YFP fluorescence accumulating in STIM1-containing puncta is reduced in cells expressing the C143A mutant. We additionally show that the mutation reduces the Pearson’s co-localization coefficient between mCh-STIM1 and ORAI1-YFP (new Figure 3F). We believe that these data provide sufficient evidence that the number of acylation-deficient ORAI1 channels recruited to STIM1-containing ER-PM junctions is reduced.

12. Replicates and quantitation of the ERK phosphorylation will be needed to conclude that ERK signaling is inhibited by C143A. Only a single gel is shown.

We have repeated the ERK phosphorylation experiments and include the quantification in the new Figure 6 supplement 3.

Reviewer #2:Orai1 contains three endogenous cysteine residues whose role in regulating channel function have been drawn much attention over the past decade. Carreras-Sureda and co-workers find that one cysteine, C143, located in transmembrane 2 can by modified via s-acylation, which impacts the activity and physiological role of Orai1 channels in T cells. PEG-5K and tritiated palmitate incorporation assays show that Orai1 is s-acylated and the site of acylation is narrowed to C143 from mutational analysis of the endogenous cysteines. The C143A mutation which should eliminate s-acylation, drastically reduced SOCE and CRAC currents independently of surface protein expression, suggesting that acylation is important for Orai1 gating and activity. Using the C143A mutant, the study shows that C143A Orai1 mobility in the plasma membrane is reduced and accumulation into STIM1 punctae following store depletion is impaired. Moreover, the mutant Orai1 appears to be excluded from cholesterol-rich lipid rafts, raising the possibility that Orai1 fails to get into zones of signaling complexes when S-acylation is blocked. Using RNA suppression and protein over-expression methods, authors also show that the protein acyltransferase enzyme, PAT20, is likely involved in S-acylating Orai1. Finally, using Jurkat T cells in which the endogenous Orai1 is CRISPered out and C143A Orai1 is expressed, the authors determine that stable transfection of C143A, but not WT Orai1, into Orai1-deficient Jurkat T cells impairs Ca signals following T cell receptor stimulation, T cell receptor assembly, NFAT signaling, and immune-synapse formation establishing the physiological relevance of s-acylation. Overall, the studies are well-done and the discovery that Orai1 is acylated is significant with potential to open up new avenues of research on Orai1 channel regulation. There are some key weaknesses which should be addressed to strengthen the conclusions.

Please refer to essential revisions. We have now performed several experiments that we believe demonstrate that S-acylation of ORAI1 is critical to sustain channel function and that this is relevant for immune synapse formation and T cell activation by TCR signalling (ERK, NFAT and IL2).

References:

Chinenov, Y. and Kerppola, T. K. (2001). Close encounters of many kinds: Fos-Jun interactions that mediate transcription regulatory specificity. *Oncogene* 20, 2438-52.

Dolmetsch, R. E., Xu, K. and Lewis, R. S. (1998). Calcium oscillations increase the efficiency and specificity of gene expression. *Nature* 392, 933-6.

Feske, S., Draeger, R., Peter, H. H., Eichmann, K. and Rao, A. (2000). The duration of nuclear residence of NFAT determines the pattern of cytokine expression in human SCID T cells. *J Immunol* 165, 297-305.

Maleth, J., Choi, S., Muallem, S. and Ahuja, M. (2014). Translocation between PI(4,5)P2-poor and PI(4,5)P2-rich microdomains during store depletion determines STIM1 conformation and Orai1 gating. *Nat Commun* 5, 5843.

McNally, B. A., Somasundaram, A., Yamashita, M. and Prakriya, M. (2012). Gated regulation of CRAC channel ion selectivity by STIM1. *Nature* 482, 241-5.

Ohno, Y., Kihara, A., Sano, T. and Igarashi, Y. (2006). Intracellular localization and tissue-specific distribution of human and yeast DHHC cysteine-rich domain-containing proteins. *Biochim Biophys Acta* 1761, 474-83.

Scrimgeour, N., Litjens, T., Ma, L., Barritt, G. J. and Rychkov, G. Y. (2009). Properties of Orai1 mediated store-operated current depend on the expression levels of STIM1 and Orai1 proteins. *J Physiol* 587, 2903-18.

Zhou, Y., Nwokonko, R. M., Cai, X., Loktionova, N. A., Abdulqadir, R., Xin, P., Niemeyer, B. A., Wang, Y., Trebak, M. and Gill, D. L. (2018). Cross-linking of Orai1 channels by STIM proteins. *Proc Natl Acad Sci U S A* 115, E3398-E3407.

[Editors’ note: further revisions were suggested prior to acceptance, as described below.]

The manuscript has been improved but there are some remaining issues that need to be addressed, as outlined below:(Comments are numbered according to the authors' rebuttal.)1a. "Assembly" has been replaced by "recruitment," which is good. But the conclusion that "acylation of Orai1 is required for recruitment and signaling at the IS" is still an overstatement. TCR is actually recruited to the IS in C143A cells, as it is enriched by a factor of ~8, about half of WT (Figure 6G). Acylation enhances TCR recruitment and signaling at the IS but is not absolutely required for either. Please correct the same conclusion on Line 54-5.

We agree and have corrected the abstract (lines 4 and 10) and main text (lines 51 and 201) to stress that acylation enhances TCR recruitment and signaling at the IS but is not absolutely required for either.

1d. See response to 1a – acylation is not required for TCR recruitment to the synapse, rather it enhances or contributes significantly to it. The text gives the impression that without acylation there is no enrichment of the TCR at the IS, which is not true.

We agree and have amended the text to stress that acylation promotes but is not required for TCR recruitment to the IS.

3. The new analysis of Orai1 fluorescence within STIM1 regions nicely addresses the comment. However, the STIM Orai1 segmentation video (from the github site) shows that the thresholding function does actually diminish the size of dim puncta relative to bright puncta. This will reduce the apparent size of the C143A Orai puncta since they are dimmer than the WT.

Thresholding indeed reduces the apparent size of the dimmer C143 Orai1 puncta. This parameter is redundant with the reduced accumulation of Orai1 fluorescence in STIM1 regions and we have removed the cluster size analysis from Figure 3. Instead, we provide additional examples to illustrate the reduced cluster size of C143 Orai1 puncta which is a prominent feature in TIRF images. We also have extended the Materials and methods section detailing the TIRF analysis (line 398).

7. The authors may keep the data in the supplemental figure, as it does seem in agreement with one previous study (Zhou et al. 2018) and no claims are made about its contribution to reduced SOCE with C143A Orai1. However, I would like to make a comment. The authors propose that the very low diffusion coefficients measured here relative to some previous values are explained by differences in the levels of expressed proteins and the geometry of the bleached regions. However, it should be noted that single particle Orai1 measurements at low expression level (Wu et al., 2014) correspond well to FRAP measurements at high expression measured with rectangular bleach (Park et al., 2009). And for STIM1, a small circular bleach area (Liou et al., 2007) gave similar values of D as a rectangular bleach (Covington et al., 2010). More importantly, the slightly slowed diffusion of C143A Orai1, while it may slightly slow down the accumulation of Orai1 in puncta, is unlikely to affect the amount of Orai1 in puncta at steady-state, which is the determinant of activity. Thus, it is not clear how the reduced diffusion rate would contribute significantly to the loss of SOCE in the mutant.

We agree that the slightly reduced diffusion rate is unlikely to affect the amount of Orai1 in puncta at steady-state. We now present these data in the supplemental figure simply to illustrate that Orai1 S-acylation does alter the mobility of the channel in membrane lipids.

10. Replotting the data in Figure 6G makes it easier to see the effect of C143A Orai1. It shows a reduction of Orai1, actin, and TCR accumulation at the synapse. However, it is important to note that these are all partial effects: all three are still enhanced at the synapse relative to the distal pole (Orai by 2-fold, actin by 3-fold, and TCR by 8-fold). Thus, it is not correct to say that acylation is "required for TCR recruitment" (line 4, 12), or that Orai C143A "fails to reach the immune synapse" (line 58-9). These kinds of absolute statements should be changed throughout the text to better reflect what the data show.

We appreciate this comment and have amended the text throughout the manuscript to avoid these kinds of absolute statements.